# Systemic metastasis-targeted nanotherapeutic reinforces tumor surgical resection and chemotherapy

Minjun Xu[1,2], Kaili Hu[3], Yipu Liu[1,2], Yukun Huang[1,2,4], Shanshan Liu[1,2], Yu Chen[1,2], Dayuan Wang[4], Songlei Zhou[1,2], Qian Zhang[4], Ni Mei[5], Huiping Lu[6], Fengan Li[1], Xiaoling Gao [4✉] & Jun Chen [1,2✉]

Failure of conventional clinical therapies such as tumor resection and chemotherapy are mainly due to the ineffective control of tumor metastasis. Metastasis consists of three steps: (i) tumor cells extravasate from the primary sites into the circulation system via epithelial-mesenchymal transition (EMT), (ii) the circulating tumor cells (CTCs) form "micro-thrombi" with platelets to evade the immune surveillance in circulation, and (iii) the CTCs colonize in the pre-metastatic niche. Here, we design a systemic metastasis-targeted nanotherapeutic (H@CaPP) composed of an anti-inflammatory agent, piceatannol, and an anti-thrombotic agent, low molecular weight heparin, to hinder the multiple steps of tumor metastasis. H@CaPP is found efficiently impeded EMT, inhibited the formation of "micro-thrombi", and prevented the development of pre-metastatic niche. When combined with surgical resection or chemotherapy, H@CaPP efficiently inhibits tumor metastasis and prolonged overall survival of tumor-bearing mice. Collectively, we provide a simple and effective systemic metastasis-targeted nanotherapeutic for combating tumor metastasis.

[1] Shanghai Pudong Hospital & Department of Pharmaceutics, School of Pharmacy, Fudan University, Shanghai, People's Republic of China. [2] Key Laboratory of Smart Drug Delivery, Ministry of Education, School of Pharmacy, Fudan University, Shanghai, People's Republic of China. [3] Murad Research Center for Modernized Chinese Medicine, Institute of Interdisciplinary Integrative Medicine Research, Shanghai University of Traditional Chinese Medicine, Shanghai, People's Republic of China. [4] Department of Pharmacology and Chemical Biology, State Key Laboratory of Oncogenes and Related Genes, Shanghai Universities Collaborative Innovation Center for Translational Medicine, Shanghai Jiao Tong University School of Medicine, Shanghai, People's Republic of China. [5] Shanghai Center for Drug Evaluation and Inspection, Shanghai, People's Republic of China. [6] Department of Pharmacy, Shanghai Pudong Hospital, Fudan University, Shanghai, People's Republic of China. ✉email: shellygao1@sjtu.edu.cn; chenjun@fudan.edu.cn

Metastasis is the foremost cause of cancer-related death[1,2]. Current clinical treatments, including tumor resection, chemotherapy, and radiation therapy, can effectively remove the primary tumor but still fail to inhibit and might even induce tumor metastasis[2,3]. Two main tumor clinical therapies, surgical resection, and chemotherapy can induce immunosuppressive and inflammatory responses, which would eventually facilitate tumor metastasis. For instance, surgical resection would induce the acute release of angiogenetic factors and immunosuppressive chemokines to promote the cell invasion by increasing tumor angiogenesis and regulatory T cells (Treg cells) cells infiltration[3]. Chemotherapies could also contribute to metastasis. Once the normal cells such as the endothelial cells and cancer-associated fibroblasts exposed to chemotherapeutics, these cells would express a high level of pro-inflammatory molecules such as interleukin-6 (IL-6) and IL-8 to recruit immunosuppressive cells and therefore lead to the suppression of anti-tumor immune response. Therefore, exploring an effective metastasis-specific therapeutic based on the underlying mechanism of tumor metastasis is critical to cancer therapy[4–6].

During the progression of metastasis, malignant tumor cells act as "seeds" to migrate away from the primary site and invade into the distant site that is termed as "soil". It includes three main steps[7–9]: invasion, circulation and the colonization phases. (i) In the invasion phase, the tumor cells escape from the primary tumor tissue to surrounding tissues and extravasate into the neighboring blood vessels. In this phase, the motility and invasiveness of tumor cells were enhanced by epithelial-mesenchymal transition (EMT)[10]. In the process of EMT, cell-cell junction proteins such as E-cadherin are downregulated, while the expression of mesenchymal proteins such as vimentin increases[11,12]. In addition, EMT takes place at the site of wound healing and is a kind of inflammatory response[11]. (ii) In the circulation phase, the tumor cells that enter and survive in the bloodstream are called circulating tumor cells (CTCs)[13,14]. During this phase, platelets are bound to CTCs, forming the special structure called "micro-thrombi", which protects CTCs from natural killer cells (NK cells) in circulation[15,16]. (iii) In the colonization phase, CTCs infiltrate and colonize in the distant organs[17]. The distant organs that provide a suitable niche for CTCs colonization are defined as the pre-metastatic niche[18–20]. Compared with normal tissues, the pre-metastatic niche shows higher expression of adhesion molecules, such as intercellular cell adhesion molecule-1 (ICAM-1) and vascular cell adhesion molecule-1 (VCAM-1)[21,22]. Besides, the increased expression of metalloproteinase-9 (MMP-9) in the niche accelerates extracellular matrix (ECM) degradation, resulting in the colonization of CTCs[23,24]. Moreover, the pre-metastatic is an inflammatory site, overexpressing S100 protein (e.g., S100A9), which would recruit myeloid-derived suppressor cells (MDSCs) such as macrophage antigen (Mac) 1+ myeloid cells to the niche, and result in CTCs' colonization in the site[13,25]. Due to the systematic processes, there is no effective clinical therapeutic targeting tumor metastasis yet. Some therapeutic strategies such as intervening EMT process via N-cadherin antibody, targeting CTCs, and hindering the formation of a pre-metastatic niche have been proposed to combat tumor metastasis[13,14,20,26–28]. However, the therapeutics targeting a single step of metastasis failed to halt the multi-steps of metastasis. When considering an effective therapeutic strategy to the clinical area, a systemic metastasis-targeted therapy that hinders the whole metastasis process might provide a more powerful therapy approach.

Herein, we designed a systemic metastasis-targeted nanotherapeutic (H@CaPP) for co-delivering an anti-inflammatory agent, piceatannol (PIC), and an anti-thrombotic agent, low molecular weight heparin (LMWH), to combat tumor metastasis by hindering the multiple steps of tumor metastasis (Fig. 1). Piceatannol is expected to effectively inhibit metastasis by suppressing signal

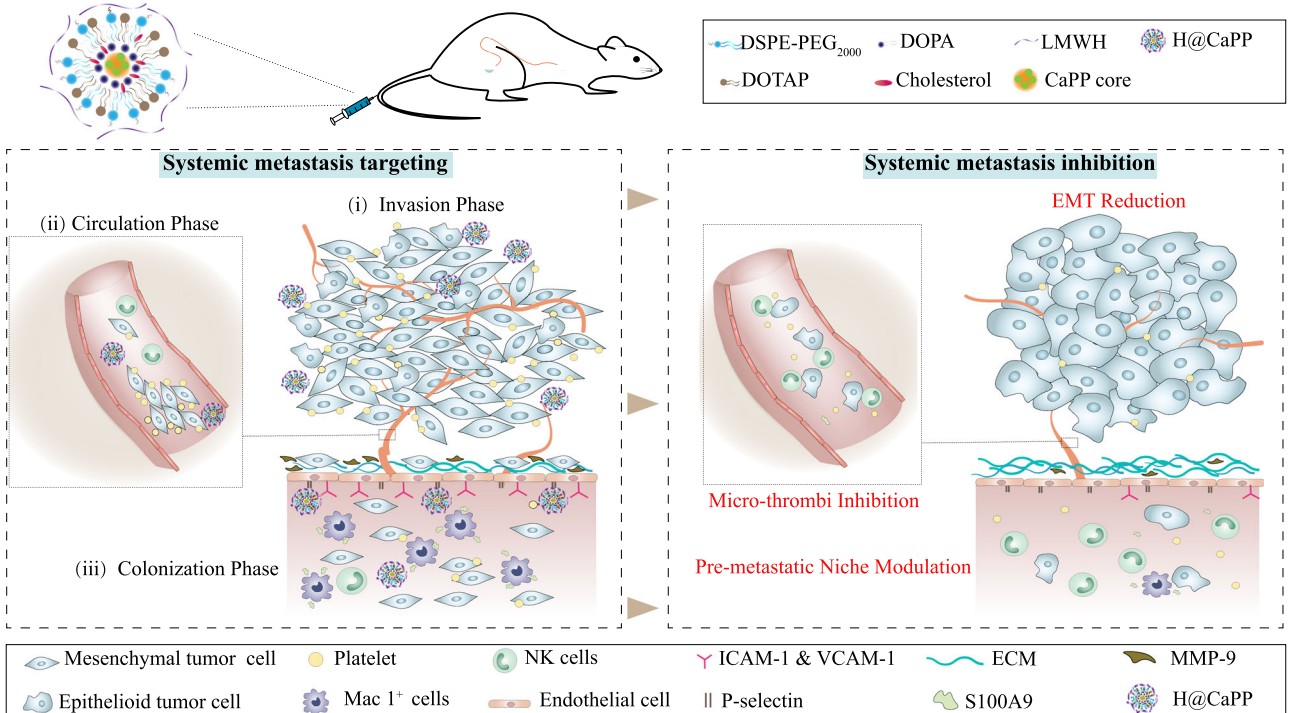

**Fig. 1 Schematic illustration of the composition and anti-metastasis function of H@CaPP.** H@CaPP was intravenously administered to suppress the whole metastasis process. H@CaPP hindered invasion phase via inhibition EMT process in the primary tumor, hindered colonization phase via inhibiting the formation of "micro-thrombi" in the circulation, and hindered the colonization phase via targeting P-selectin in the pre-metastatic niche and inhibiting typical aberrant alterations in the niche.

transducers and activators of transcriptionnuclear-3 (STAT-3) and nuclear factor-κB (NF-κB), etc[29,30]. LMWH, a class of anticoagulant medication commonly used in clinic, possesses a high affinity to P-selectin that is overexpressed on both activated endothelial cells (ECs) in the pre-metastatic niche and activated platelets[18–22,31,32]. Therefore, LMWH may serve as not only a targeting ligand to P-selectin but also an anti-micro-thrombi component. Nanotechnology holds plentiful advantages in drugs co-delivery, including improving stability and circulation, and especially, integrating different agents into one platform. Accordingly, here a calcium-phosphate liposome (CaP)-based nanostructure was developed for targeting and hindering the multiple steps of tumor metastasis through the co-delivery of piceatannol and LMWH. For effectively co-delivering and improving drug-loading efficiency, piceatannol phosphate (PP) was synthesized through the phosphorylation of the hydroxyl groups of PIC. PP was then precipitated with calcium to form calcium phosphate nanoparticles (CaPP), which were further decorated with LMWH via electrostatic interaction on the surface to form a systemic metastasis-targeted nanotherapeutic (H@CaPP). As expected, H@CaPP successfully impeded the multiple steps of metastasis: inhibiting the EMT process in the invasion phase; hindering the formation of "micro-thrombi" in the colonization phase; suppressing pre-metastatic niche via binding to P-selectin, decreasing the adhesiveness of ECs, reversing the remodeling of extracellular matrix, and hindering the development of the immunosuppressive inflammatory site (Fig. 1). When combined with chemotherapy or surgical resection, H@CaPP significantly reduced lung tumor metastasis and prolonged the overall survival of mice bearing 4T1 orthotropic mammary tumor. As proof of concept, this study combined anti-inflammatory drugs and anticoagulants into a targeted nanotherapeutic, providing a safe and potential strategy for adjuvant therapy of metastatic tumors.

## Results

**Preparation and characterization of H@CaPP.** H@CaPP was prepared through the procedure as demonstrated in Fig. 2a. Piceatannol prodrug (PP) was synthesized to improve its water solubility and facilitated its encapsulation in H@CaPP (Supplementary Fig. 1). PP-loaded nano-sized calcium phosphate was prepared and coated with dioleoylphosphatydic acid (DOPA) through the water-in-oil microemulsion method[33–36]. Afterward, the inner leaflet lipid was encapsulated with distearoyl-*sn*-glycero-3-phosphoethanolamine-*N*-[maleimide(polyethyleneglycol)-2000] (DSPE-PEG2000), cholesterol, and 1,2-dioleoyl-3-trimethylammonium-propane (DOTAP) to form the positively charged lipid nanoparticles named CaPP[37–39]. Finally, the negatively charged LWMH was coated on CaPP via electrostatic interaction to obtain H@CaPP. With a similar method, D@CaPP without targeting capability was prepared by replacing LMWH with dextran to serve as the control.

The chemical structure of PP was confirmed by NMR analysis (Supplementary Figs. 2–9). H@CaPP showed a uniform size of 76 nm (Fig. 2b–d) and the zeta potential is approximately −37 mV (Fig. 2e). PP was effectively encapsulated with high loading capacity (8.05% ± 0.54%) and encapsulation efficiency (64.81% ± 4.19%). The binding efficiency of LMWH was 68.24% ± 5.04%. Transmission microscopy (TEM) analysis found a dimmer LMWH layer (around 15 nm) on the surface of H@CaPP, which was not found on that of CaPP (Fig. 2d and Supplementary Fig. 10). These data demonstrated that PP was successfully loaded into the nanoparticles and LMWH was successfully bound on the surface of H@CaPP. The release profiles of PP were determined at 37 °C in phosphate buffer solution (PBS, pH 7.4). PP showed a

burst release in the first 2 h and reached a plateau at 4 h (Fig. 2f) in the PBS group. In contrast, <25% of PP was released from H@CaPP and D@CaPP after 24 h incubation. The data showed H@CaPP exhibited negligible changes in the size and zeta potential in PBS and slight changes in 10% FBS within 24 h (Fig. 2g, h). In order to investigate the stability of LMWH on the surface of H@CaPP, we evaluated the LMWH release from nanotherapeutics in PBS. As shown in Supplementary Fig. 11, LMWH showed a burst release in the first 2 h and reached a plateau at 4 h. In contrast, less than 16% of LMWH was released from H@CaPP after 24 h incubation. According to the results, the heparin on H@CaPP achieved great stability in the circulation. Then, we performed the experiment to determine the profiles of PP release from H@CaPP at the intracellular conditions. As reported, the calcium-phosphate liposome would be internalized by endosomal/lysosomal pathway[40–42]. Therefore, we evaluated the PP release from H@CaPP in PBS at pH 6.5 (mimicking endosomes) and 5.0 (mimicking lysosomes). In pH of lysosomes, the H@CaPP released 83.0% of PP in 24 h. Whereas, in pH of endosomes, only 35.5% of PP was released from the H@CaPP in 24 h (Supplementary Fig. 12). The result indicated that the release rate of PP was significantly dependent on the pH value and the drug was released from lysosomes into the cytoplasm. To study the intracellular conversion of PP to piceatannol, we incubated 4T1 and HUVECs cells with H@CaPP for 4 h, and successfully detected piceatannol in the culture media, which indicated that PP could be transformed to the active drug piceatannol in cells (Supplementary Fig. 13).

Furthermore, we performed an MTT assay to evaluate the toxicity of H@CaPP in vitro[43]. For the analysis, HUVECs and 4T1 cells were treated with H@CaPP at the PP concentration ranging from 1 to 30 μM. The viability of both types of the cells was higher than 80% even when the concentration of PP or piceatannol was up to 30 μM (Supplementary Fig. 14a, b), suggesting that H@CaPP exhibited no significant cytotoxicity. Then, we performed the hemolytic assay to investigate the blood safety of H@CaPP[24,44]. The abs value of H@CaPP did not show a significant difference with that of the negative control (Supplementary Fig. 15), suggesting that H@CaPP possesses negligible hemolytic toxicity.

**H@CaPP targeted primary tumor and pre-metastatic site.** For achieving systemic therapy of tumor metastasis, H@CaPP would target both primary tumor and pre-metastatic site. Therefore, we evaluated the targeting efficacy of H@CaPP. Firstly, the primary tumor cells could induce the spontaneous formation of pre-metastatic niches. Therefore, in order to identify the period of pre-metastatic niche formation, the lungs of tumor-bearing mice were resected and stained with hematoxylin-eosin staining (H&E) every second day to observe lung morphology. From days 4 to 14 after tumor inoculation, changes in terms of lung morphology were observed in the tumor-bearing mice, with a decrease in the number of alveoli and an increase in the number of inflammatory cells. And the initial visible lung metastatic areas were observed on the 16th day after inoculation (Supplementary Fig. 16). To verify the formation of the pre-metastatic niche in the lung, the expression S100A9 and MMP-9, two marker proteins of the pre-metastatic niche, were tested through immunohistochemistry (IHC)[13,20,23–25]. Compared with the normal mice group, lung section from tumor-bearing mice exhibited enhanced S100A9 and MMP-9 expression on the 14th day (Supplementary Fig. 16b). Thus, the period from days 4 to 14 was defined as the period of pre-metastatic niche formation.

Then the targeting capabilities of H@CaPP for both primary tumor and pre-metastasis were further investigated

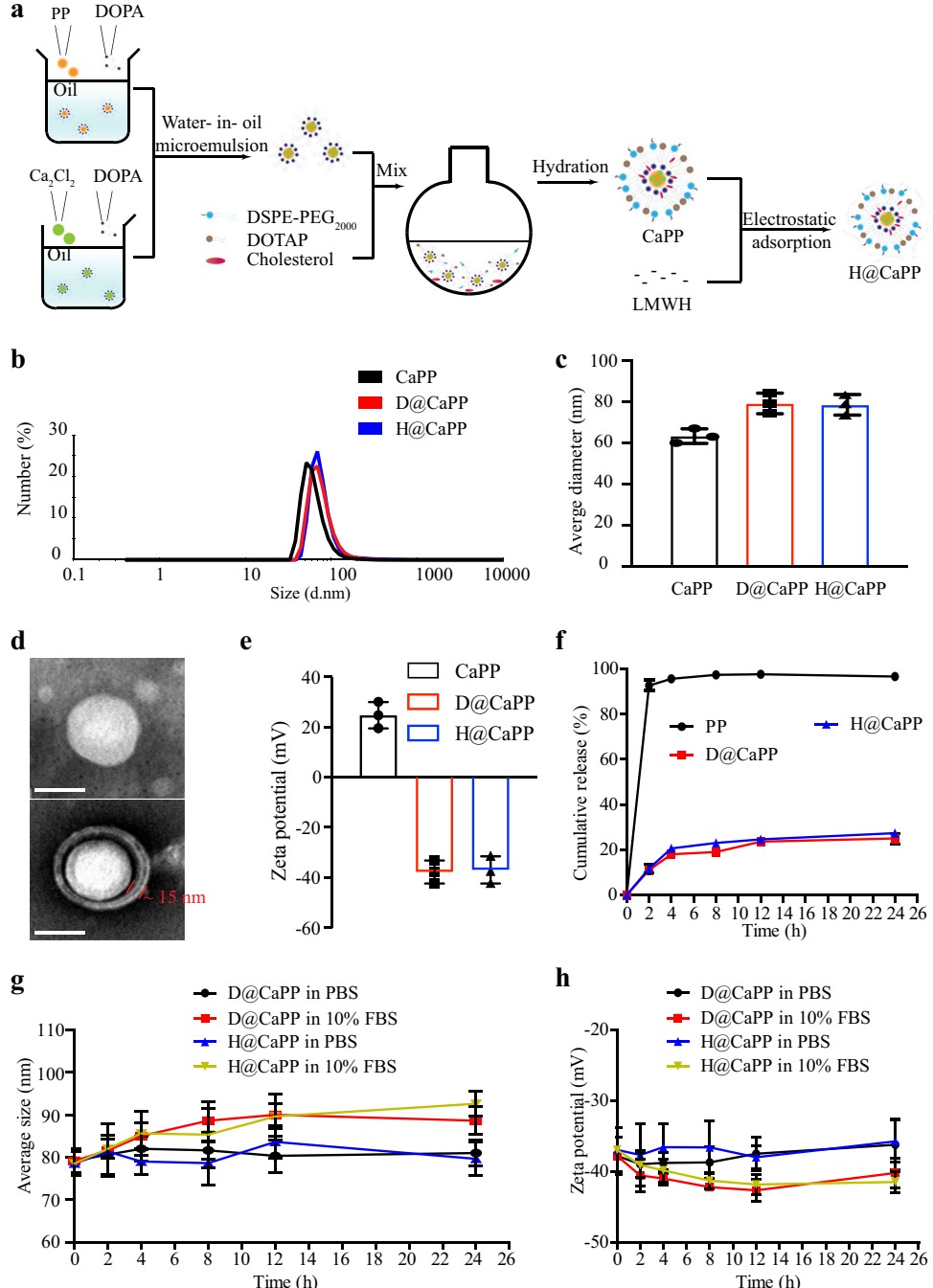

**Fig. 2 Preparation and characterization of H@CaPP. a** Schematic illustration of the preparation route of H@CaPP. **b** Particle size of CaPP, D@CaPP, and H@CaPP measured by dynamic light scattering. **c** Average diameter of CaPP, D@CaPP, and H@CaPP (mean ± SD, $n = 3$ samples per group). **d** Morphology of CaPP and H@CaPP. Scale bar, 50 nm. The experiments were repeated three times independently. **e** Surface zeta potential of CaPP, D@CaPP, and H@CaPP (mean ± SD, $n = 3$ samples per group). **f** The PP release profile of D@CaPP, H@CaPP, or PP in PBS (pH 7.4) at 37 °C with gentle stirring (mean ± SD, $n = 3$ samples per group). **g** Average size of D@CaPP and H@CaPP within 24 h incubation with 10% FBS or PBS (pH 7.4) at 37 °C, respectively. **h** Zeta potential of D@CaPP and H@CaPP incubated with 10% FBS or PBS (pH 7.4) at 37 °C, respectively within 24 h (mean ± SD, $n = 3$ samples per group). Error bars represent SD.

in vivo[14,45–47]. The DiR-labeled nanoparticles were intravenously administrated and the time-dependent biodistribution was recorded via the IVIS instrument (Fig. 3a). In the primary tumors, DiR-loaded D@CaPP and H@CaPP achieved 5.42-fold and 6.81-fold higher tumor accumulation than DiR, respectively (Fig. 3c, d). In the pre-metastatic sites, the in vivo fluorescence and ex vivo semi-quantitative analysis of lung at 24 h illustrated that the accumulation of H@CaPP was 3.65-fold and 2.15-fold

higher than that of free DiR and D@CaPP, respectively (Fig. 3b–d). These data suggested that H@CaPP could not only accumulate at the primary tumor sites but also effectively target the pre-metastatic niche in vivo.

In order to investigate the underlying mechanisms of the targeting activity of H@CaPP, HUVECs pre-treated with tumor necrosis factor-α (TNF-α) overexpressed were served as an in vitro model[13]. We measured the expression of two common

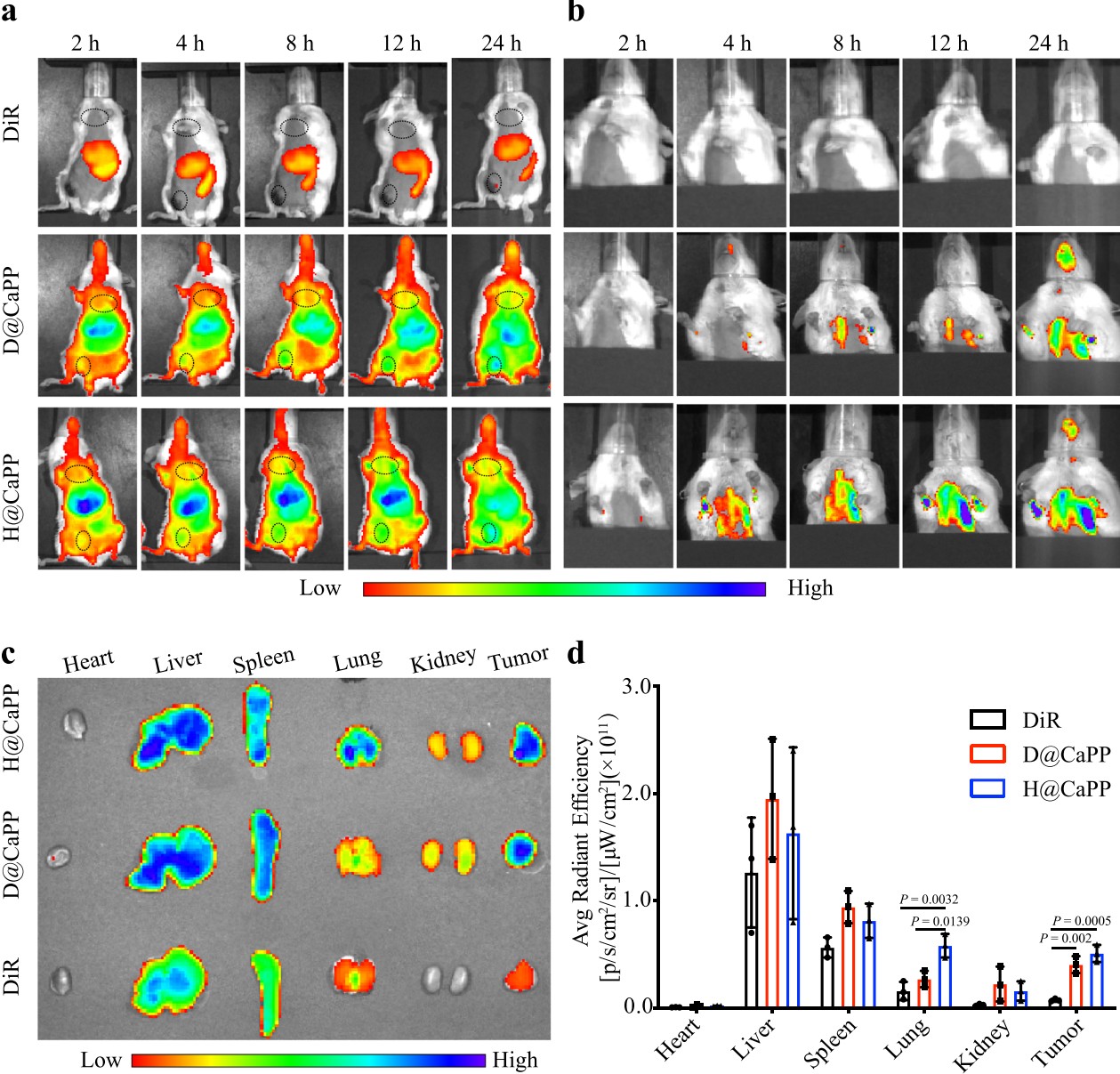

**Fig. 3 H@CaPP targeted primary tumor and pre-metastatic site in vivo. a** In vivo fluorescence imaging of the whole body of orthotropic tumor-bearing mice at 2, 4, 8, 12, and 24 h after intravenous injection of free DiR, DiR-labeled D@CaPP, and H@CaPP at the dose of 1 mg/kg DiR. The black cycles indicate the sites of tumors and lungs. **b** Fluorescence imaging of lung of tumor-bearing mice at 2, 4, 8, 12, and 24 h after intravenous injection with free DiR-labeled D@CaPP and H@CaPP at the dose of 1 mg/kg DiR. Black cardboards were utilized to block fluorescence signals from tumors and other organs. **c** Images of the organs and tumors of mice in each group at 24 h. **d** Semi-quantitative analysis of fluorescence intensity of the ex vivo organs and tumors (mean ± SD, $n$ = 3 mice per group). One-way ANOVA with Tukey's multiple comparisons test (one-sided) was used for **d**. Error bars represent SD.

metastatic-related selectins, P-selectin and E-selectin, on the surface of HUVECs. As shown in Supplementary Fig. 17, HUVECs expressed more P-selectin and E-selectin after stimulated with TNF-α, consistent with the observation from the pre-metastatic model in vivo. Furthermore, we investigated the pre-metastatic niche targeting effect of H@CaPP via a cellular uptake assay. In HUVECs pre-treated with TNF-α, the uptake of H@CaPP significantly increased by 3.30-fold compared with that of D@CaPP. The blockage of P-selectin led to a significant decrease of 49.03% of H@CaPP uptake efficiency in HUVECs pre-treated with TNF-α, whereas there was no obvious change in the E-selectin blockage group (Fig. 4a, d). Subsequently, we studied the binding capability of H@CaPP to P-selectin in vivo. The tumor-bearing mice and normal mice were intravenously

injected with DiI-labeled nanoparticles. In the normal mice, none of D@CaPP and H@CaPP showed specific accumulation in the lungs because of the low P-selectin expression therein. In contrast, in the tumor-bearing mice, compared with that of CaPP and D@CaPP, a much higher accumulation of H@CaPP was observed in the lung along with the enhanced P-selectin levels (Fig. 4c). In addition, as a large number of platelets and ECs are activated in the tumor microenvironment by endogenous cytokines, P-selectin is overexpressed at primary tumor[20,48–50]. Hence, H@CaPP could efficiently target the pre-metastasis niche and the primary tumor.

**H@CaPP hindered invasion of the primary tumor via EMT inhibition**. The invasion of tumor cells is an initial and essential

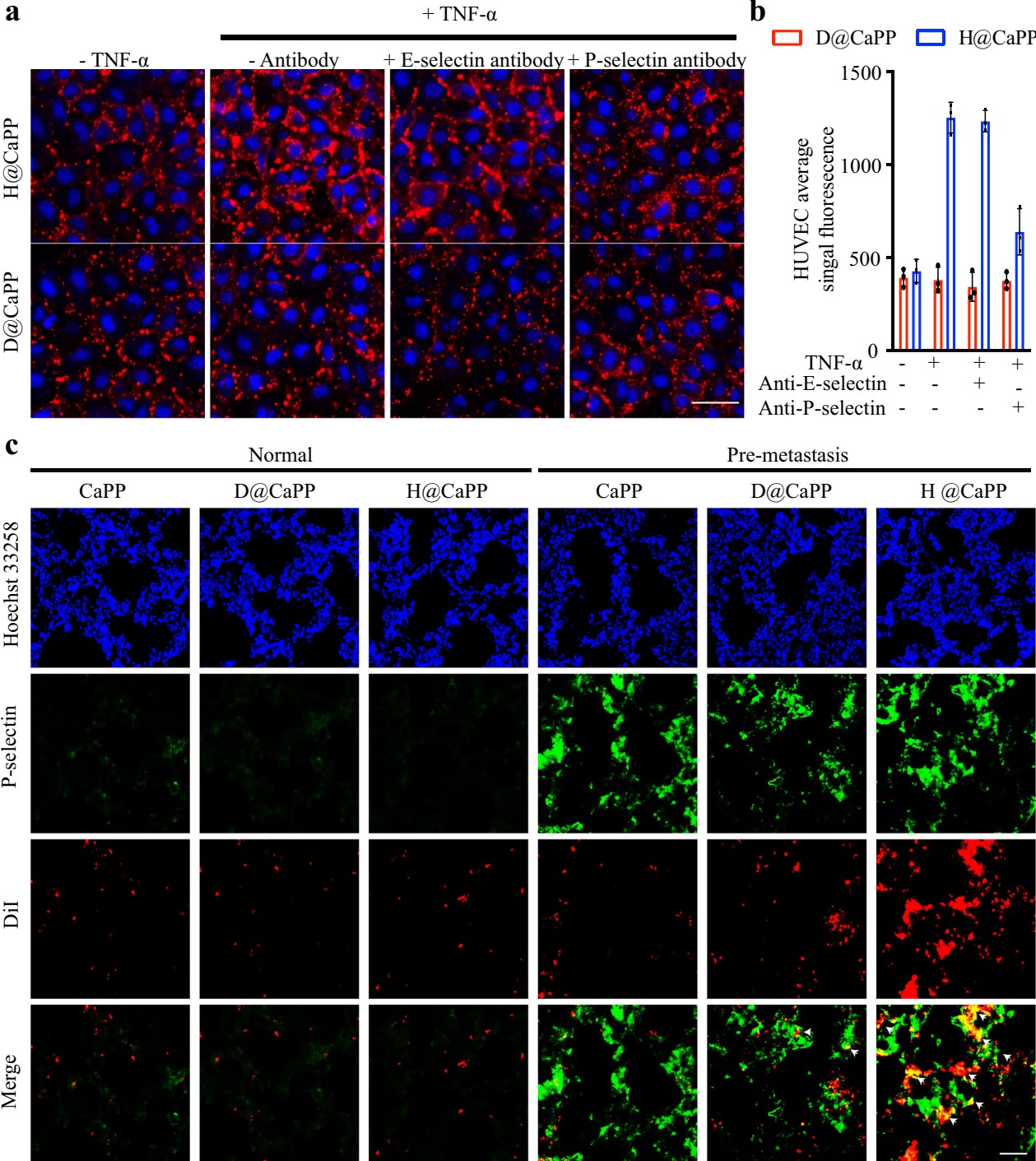

**Fig. 4 H@CaPP targeted the pre-metastatic niche. a** Uptake of D@CaPP and H@CaPP in HUVECs. DiI-labeled D@CaPP and H@CaPP (red) were incubated with a HUVECs pre-incubated with or without TNF-α (50 ng/mL, 4 h), respectively. In order to investigate the mechanism, an anti-P-selectin antibody was incubated with HUVECs for 2 h before adding nanotherapeutics. Afterward, HUVECs on a 6-wells plate was imaged after staining nuclei by Hoechst 33258 (blue) for eight minutes. Scale, bar, 20 μm. **b** Quantification of HUVECs uptake of nanotherapeutics under various conditions (ANOVA, n.s. indicated no significant difference, mean ± SD, $n = 3$ images per group). **c** Expression of P-selectin in mice lungs examined by immunofluorescent staining. Normal mice and tumor-bearing mice were intravenously injected with DiI-labeled nanotherapeutics (1 mg/kg DiI), respectively. Blue: Hoechst 33258 stained nuclei; green: P-selectin pre-metastatic sites; red: DiI-labeled nanoparticles. Scale bar, 50 μm. The experiments were repeated three times independently. Error bars represent SD.

step for metastasis. EMT, an inflammatory process, is commonly regarded as an initiator of tumor cells invasion[11,12,51]. For this reason, we next investigate whether the nanoparticles could hinder tumor cell invasion by inhibiting EMT. For the analysis, the number of 4T1 cells that penetrated the Matrigel-coated

polycarbonate membrane in the piceatannol groups was 60.4% of that in the PBS group. D@CaPP and H@CaPP groups exhibited strong invasion-inhibition abilities (Fig. 5a, c). Then, two typical markers of EMT, E-cadherin, and vimentin, were detected by Western blot in vitro and immunohistochemistry of tumor tissue

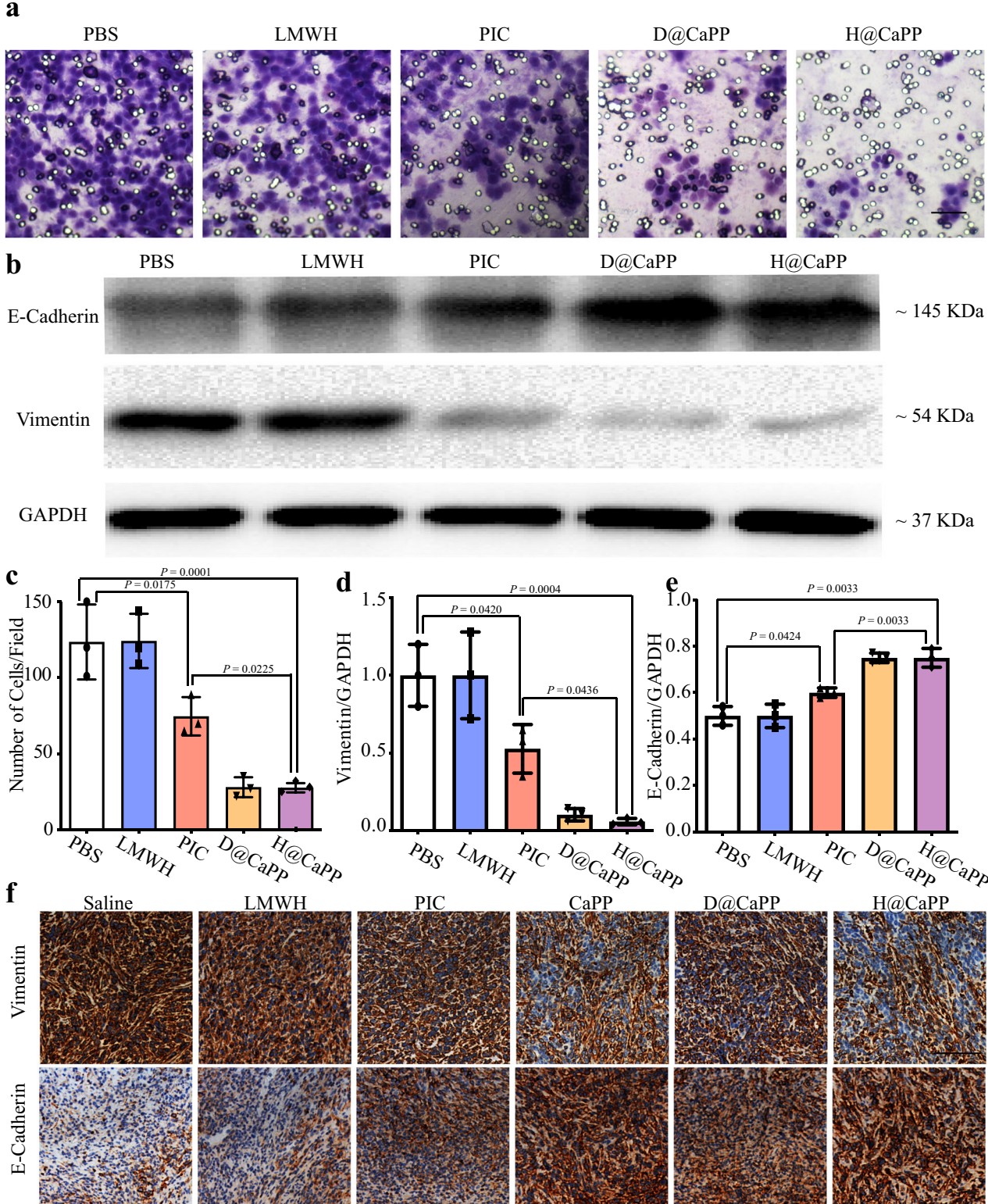

**Fig. 5 H@CaPP hindered invasion of the primary tumor by inhibiting the EMT process. a** Images and **c** quantitative analyses of the number of invaded 4T1 cells after incubating tumor cells with PBS, LMWH, piceatannol, D@CaPP, or H@CaPP (dose of free piceatannol or PP at 10 µM, dose of Dextran or LMWH at 1.3 mg/mL) for 24 h. The cells were stained with crystal violet and each purple spot represented one invaded cell at the field of vision (ANOVA, means ± SD, *n* = 5 fields per group). **b** Representative image of Western blot analysis of Vimentin and E-cadherin in 4T1 cells. **d** Quantitative analysis of the expression of Vimentin and **e** E-cadherin in 4T1 cells (ANOVA, means ± SD, *n* = 3 samples per group). **f** Representative images of immunological analysis of Vimentin and E-cadherin (brown) in tumors after intravenously administrated with saline, LMWH, piceatannol, CaPP, D@CaPP, and H@CaPP (i.v. at a dose of piceatannol or PP at 0.020 mmol/kg, at a dose of Dextran or LMWH at 10 mg/kg) for seven times in 14 days. Scale bar, 50 µm. One-way ANOVA with Tukey's multiple comparisons test (one-sided) was used for **c**–**e**. Error bars represent SD.

section in vivo[10]. High expression of vimentin, a mesenchymal-like cell-cell adhesion molecule, can lead to increasing invasiveness of tumor cells. As for E-cadherin, an epithelial-like cell-cell adhesion molecule, its reduction indicates the potential of metastasis. The expression of E-cadherin in 4T1 cells treated by piceatannol, D@CaPP, or H@CaPP was increased by around 21%, 44%, and 45% for 24 h, compared to PBS group, respectively; and the expression of vimentin was reduced in these groups in vitro (Fig. 5b, d, e). Consistently, as the results are shown in Fig. 5f and Supplementary Fig. 18, treatments displayed less vimentin expression and more E-cadherin expression were observed in H@CaPP-treated groups than other groups in vivo ($P < 0.0001$). These results demonstrated that both piceatannol and nanoparticles could inhibit EMT and invasion of tumor cells in vitro, while the nanoparticle would enable higher accumulation of the piceatannol at tumor sites and less degradation in circulation to achieve more inhibition of the EMT process in vivo. Collectively, H@CaPP might efficiently hinder tumor cell invasion via EMT inhibition.

**H@CaPP hindered the circulation phase by inhibiting the formation of "micro-thrombi".** The massive level of platelets is significantly related to tumor metastasis in the circulation phase[16]. Platelets could aggregate on tumor cells, forming "micro-thrombi". Briefly, to form "micro-thrombi", the platelets become activated by CD 44 and integrin $\alpha IIb\beta 3$ on the surface of tumor cells[52,53]. Then, the activated platelets produced coagulation factors such as coagulation factor III to catalyze prothrombin into thrombin. Finally, thrombin contributes to platelets aberrant aggregation on the tumor cells and form the "micro-thrombi"[15,16]. The "micro-thrombi" could disguise tumor cells and prevent the cells elimination from the immune system[54,55]. LMWH is able to inhibit the aberrant aggregation of platelets by hindering the conversion of prothrombin to thrombin via promoting antithrombin III (AT) activity[31,32,56]. Therefore, we hypothesized that H@CaPP coated with LMWH could hinder the circulation phase by downregulating the formation of "micro-thrombi". Then we investigated whether H@CaPP could inhibit the adhesion of platelets to tumor cells. In vitro, the adhesion effect of platelets labeled with calcein-AM to 4T1 cells was clearly visible via confocal microscopy. Compared with the PBS group, the fluorescence intensity of platelets in the LMWH group and H@CaPP group were significantly decreased by 71.51% and 70.99%, respectively, suggesting H@CaPP could hinder the adhesion of platelets to tumor cells (Fig. 6a, b). CD-41, known as platelet glycoprotein, is a 143 kDa membrane protein and is involved in blood coagulation by mediating platelet aggregation that considered to be one of the markers of platelets. Thus, we labeled platelets with CD-41. The IHC assay showed that the platelet aggregation was less organized in H@CaPP group than other groups along with the decrease of CD-41 (Fig. 6a). These data supported that H@CaPP could interfere with the formation of "micro-thrombi" in vivo. Collectively, these results supported that H@CaPP could hinder the formation of "micro-thrombi", providing an inhibition effect in the circulation phase in tumor metastasis. Thus, H@CaPP could hinder the circulation phase via inhibiting the formation of "micro-thrombi" in vitro and in vivo.

**H@CaPP hindered the colonization phase via inhibiting the development of pre-metastatic niche.** In the colonization phase, the development of a pre-metastatic niche is a specialized microenvironment that allowed the colonization of circulating tumor cells (CTCs). The adhesion of CTCs to ECs in a pre-metastatic niche is essential for metastasis formation[7,20]. Thus, we investigated whether H@CaPP could hinder the adhesion of

tumor cells to activated ECs in the pre-metastatic site. HUVECs were pre-treated with TNF-$\alpha$ to mimic the activated ECs in the niche[13]. The number of 4T1 cells that adhered to TNF-$\alpha$ + H@CaPP-treated HUVEC monolayer was significantly reduced compared with those in other groups (Fig. 7a, b). Furthermore, to explore the mechanism, we studied the expression of two main cell adhesion molecules (ICAM-1 and VCAM-1) on activated ECs[21,22]. VCAM-1 fluorescent intensity in TNF-$\alpha$ + H@CaPP resulted in a significant reduction compared with the intensities in PBS, LMWH, piceatannol, and D@CaPP, respectively. At the same time, the fluorescent intensity of ICAM-1 demonstrated a similar trend (Fig. 7d and Supplementary Fig. 19). H@CaPP could inhibit aberrant expressions of ICAM-1 and VCAM-1 on activated ECs. The expressions of ICAM-1 and VCAM-1 in activated ECs were positively correlated with the adhesion of CTCs to activated ECs, suggesting that H@CaPP could efficiently hinder the interactions between CTCs and ECs. Besides, the increased expression of MMP-9 could result in lung tissue remodeling by degrading ECM and accelerate the migration of CTCs into the tissue[23]. In addition, S100A9 could recruit MDSCs, leading to promote the formation of the inflammatory immune niche[20,25]. Herein, we identified a high level of these four proteins as molecular markers for the development of pre-metastatic niches in vivo. As shown in Fig. 8 and Supplementary Fig. 20, immunohistochemistry assay showed that only H@CaPP significantly exhibited inhibitory effects of ICAM-1, VCAM-1, S100A9, and MMP-9 compared with other groups. Then, in order to investigate how H@CaPP affects the inflammation site, we studied the recruitment of granulocytic myeloid-derived suppressor cells (G-MDSCs) in the lung. The G-MDSCs could be recruited to inflammation sites, such as pre-metastatic niche, by multiple factors (VEGF, CXCL5, GM-CSF, IL-6, etc.) and can facilitate metastasis by causing increased vascular permeability and immunosuppression[18,57,58]. Ly6G is a marker of G-MDSC. Therefore, the expression Ly6G in the lung was positively correlated with the recruitment of G-MDSCs to the pre-metastatic niche[59]. As shown in Supplementary Fig. 21, H@CaPP significantly exhibited inhibitory effects of Ly6G compared with other groups ($P < 0.0001$). The result supported H@CaPP could hinder the recruitment of G-MDSCs and showed the anti-inflammation effect in the pre-metastatic niche. Taken together, these data supported the hypothesis that H@CaPP could inhibit pre-metastatic niche formation. Thus, H@CaPP could efficiently hinder the colonization phase via inhibiting the development of a pre-metastatic niche.

**Therapeutic effects of H@CaPP combined with chemotherapy in orthotropic primary tumor-bearing mice.** To determine whether the systemic targeted nanotherapeutic can serve as a potential adjuvant therapy to chemotherapy, the therapeutic effect of H@CaPP combined with chemotherapy (free PTX) was investigated in 4T1 tumor-bearing mice. Compared with those groups without PTX treatment, the mice treated with PTX showed a significant reduction in primary tumor growth (Supplementary Fig. 22). To evaluate the anti-metastasis effect of combined treatments, we counted the number of lung metastases nodules in lung sections[60,61]. The mean metastatic nodules in the lungs of mice treated with saline, PTX, PTX + LMWH, PTX + PIC, PTX + CaPP, PTX + D@CaPP, PTX + H@CaPP were 30, 29, 20, 23, 17, 12, and 3, respectively. Similarly, a metastatic area in the mice treated with the PTX + H@CaPP was also the smallest (Fig. 9a, b). Next, the survival of mice was investigated to evaluate the therapeutic effect of H@CaPP in combination with PTX or surgical resection. In Fig. 9c and Supplementary Table 1, the mice treated with PTX + H@CaPP (53 days) exhibited

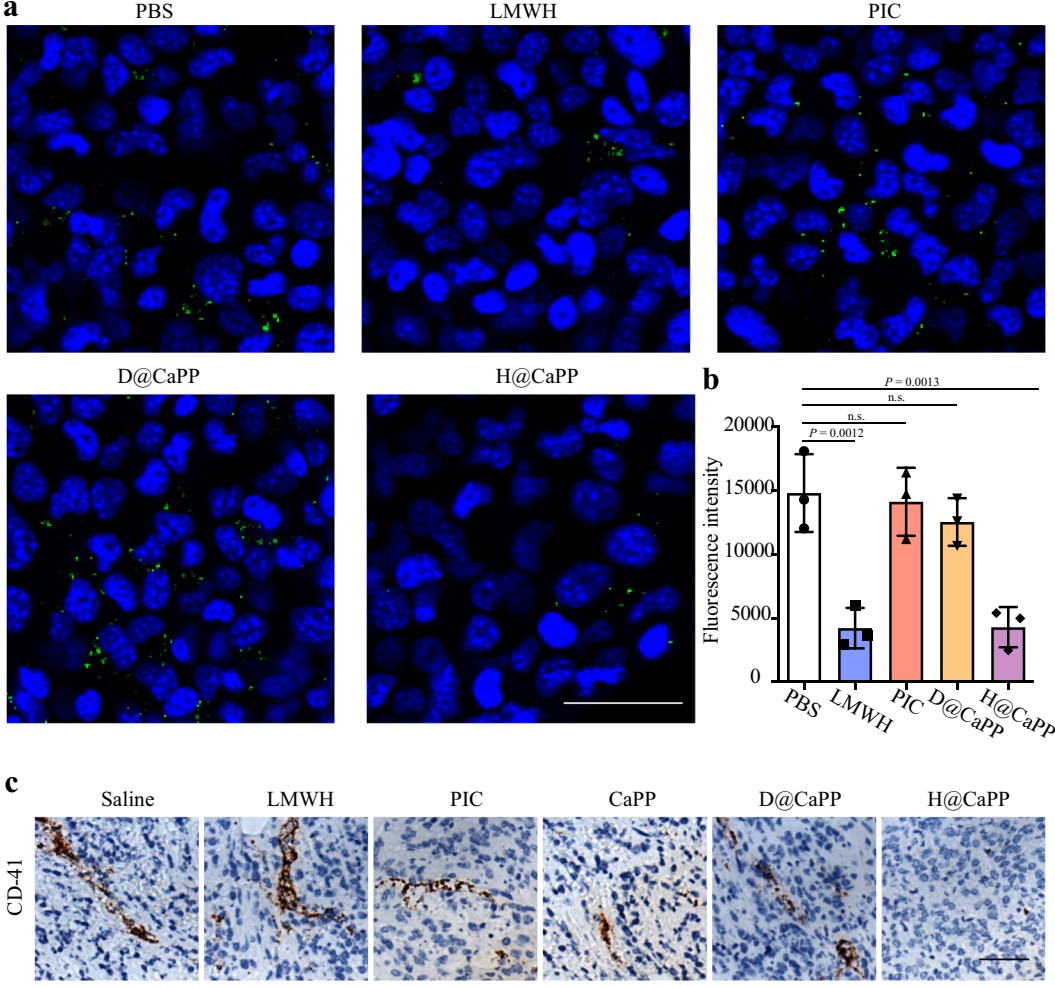

**Fig. 6 H@CaPP inhibited the formation of "micro-thrombi" by inhibiting the adhesion of platelets to tumor cells. a** Adhesion of platelets to 4T1 cells, and the inhibitory effect of PBS, LMWH, piceatannol, D@CaPP, and H@CaPP (dose of free piceatannol or PP at 10 μM, dose of Dextran or LMWH at 1.3 mg/mL). The nuclei of 4T1 cells were stained with Hoechst 33258, and platelets were obtained and labeled with calcein-AM (green). Scale bar, 50 μm. **b** The fluorescence intensity analysis of platelets adhering to 4T1 cells (n.s. indicates no significant difference with that of PBS, ANOVA, mean ± SD, $n = 3$ fields per group). **c** IHC analysis of CD-41 in tumors after intravenously administrated with saline, LMWH, piceatannol, D@CaPP, and H@CaPP (i.v. at a dose of piceatannol or PP at 0.020 mmol/kg, at a dose of Dextran or LMWH at 10 mg/kg) for seven times in 14 days. Scale bar, 50 μm. One-way ANOVA with Tukey's multiple comparisons test (one-sided) was used for **b**. Error bars represent SD.

significantly longer median survival days compared with saline (38 days), PTX (40 days), PTX + LMWH (42 days), PTX + piceatannol (40.5 days), PTX + CaPP (44 days) or PTX + D@CaPP (44.5 days). The results of the anti-metastasis effect and survival experiments demonstrated that H@CaPP combined with PTX achieved better comprehensive therapeutic efficacy towards the triple-negative breast cancer (TNBC) which is a highly metastatic tumor.

**Therapeutic effects of H@CaPP combined with surgery resection in orthotropic primary tumor-bearing mice.** To demonstrate whether H@CaPP can serve as a potential adjunct therapy against post-surgery tumor metastasis, we continued to evaluate the therapeutic effect of the nanotherapeutic combined with surgery resection in 4T1 tumor-bearing mice. Before combination with nanotherapeutic, it is vital to determine the optimal operational time. Actually, it is easier to find a proper site for surgery when the tumor is large. But the risks of mortality for the mice would be dramatically increased if it was carried out too late due to excessive surgery wounds. Therefore, we performed sequential surgical resection at a different scheduled time. Only two mice

survived after surgery on the 14th day after the tumor cells were inoculated into the fat pat (Supplementary Fig. 23). Thus, we determined the appropriate time for a surgical procedure on the 8th day once the tumor tissue could be observed and removed. Although it may increase the difficulty of surgery, surgical death was avoided. The mean metastatic nodules of lungs in saline, surgery + saline, surgery + LMWH and surgery + PIC, the surgery + CaPP, the surgery + D@CaPP and surgery + H@CaPP group was 33, 29, 16, 26, 11, 10, and 2, respectively (Fig. 10a, b). Consistently, the survival analysis showed that the surgery + H@CaPP group displayed the longest medium survival time (66.5 days), while the medium survival time of saline, surgery + saline, surgery + LMWH and surgery + piceatannol, the surgery + CaPP, the surgery + D@CaPP and surgery + H@CaPP group was 38, 49.5, 52.5, 54.5, 52, and 56.5 days, respectively (Fig. 10c and Supplementary Table 2). The results demonstrated that H@CaPP combined surgery resection achieved better anti-metastasis therapeutic efficacy in TNBC.

**Safety evaluation in vivo.** For testing whether the nanotherapeutic is a safe therapy, its toxicity was assessed in the animals

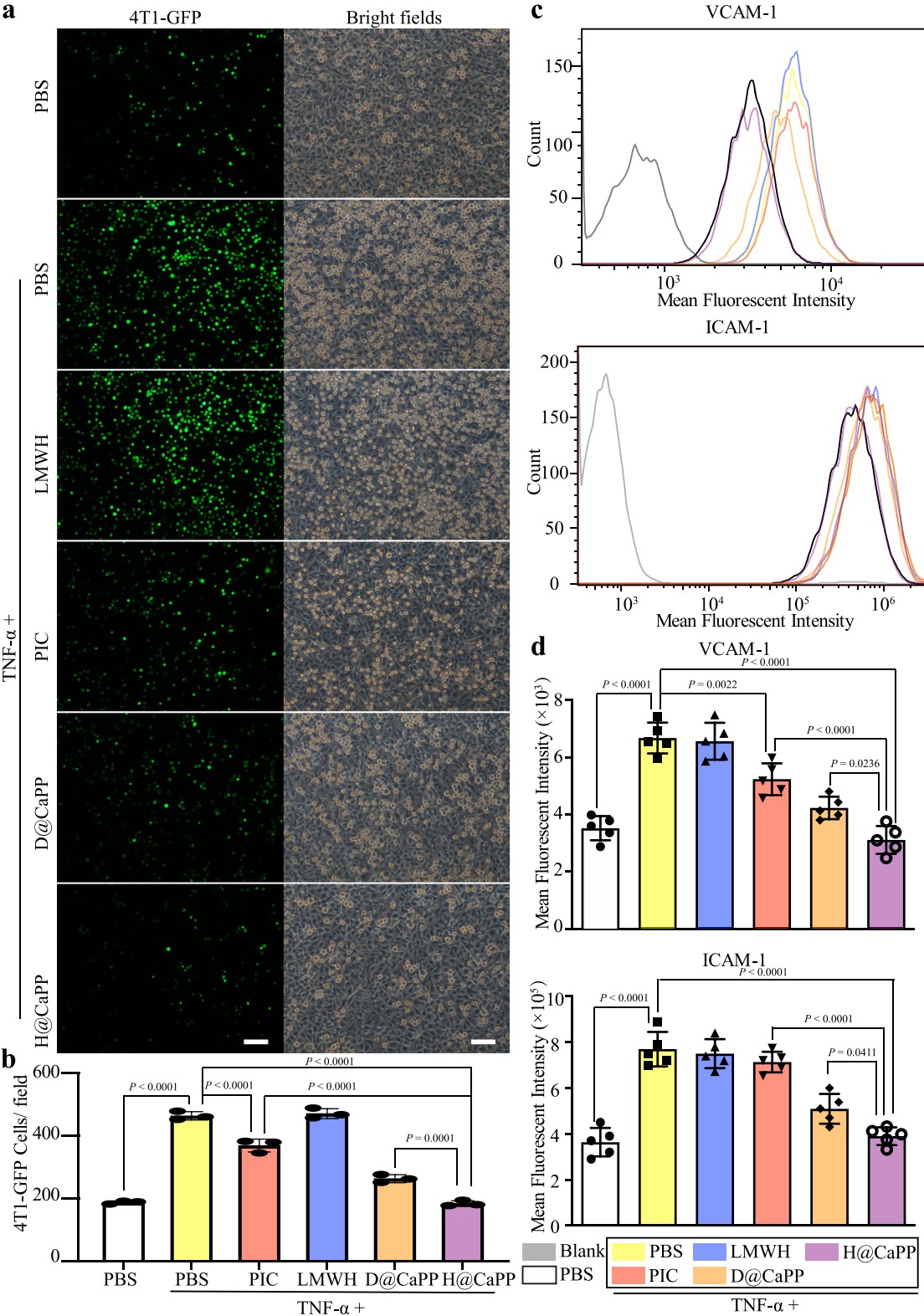

**Fig. 7 Evaluation of the adhesion of ECs in the pre-metastatic site in vitro. a** Images of 4T1 cells (labeled calcein-AM) adhered to HUVECs which were treated with PBS, TNF-α + PBS, TNF-α + LMWH, TNF-α + piceatannol, TNF-α + D@CaPP, and TNF-α + H@CaPP (TNF-α = 50 ng/mL, a dose of free piceatannol or PP at 10 μM, dose of Dextran or LMWH at 1.3 mg/mL) for 24 h, respectively. Left photo, fluorescent field; right photo, bright field. Scale bar, 50 μm. **b** Number of 4T1 cells in per field from each group (ANOVA, mean ± SD, n = 3 fields per group). **c** Flow cytometry analysis of VCAM-1 and ICAM-1 on HUVECs treated as **a**, respectively. Cells without the incubation of antibodies were served as blank control. **d** The fluorescence intensity of VCAM-1 and ICAM-1 in each group (ANOVA, mean ± SD, n = 5 samples per group). One-way ANOVA with Tukey's multiple comparisons test (one-sided) was used for **b** and **d**. Error bars represent SD.

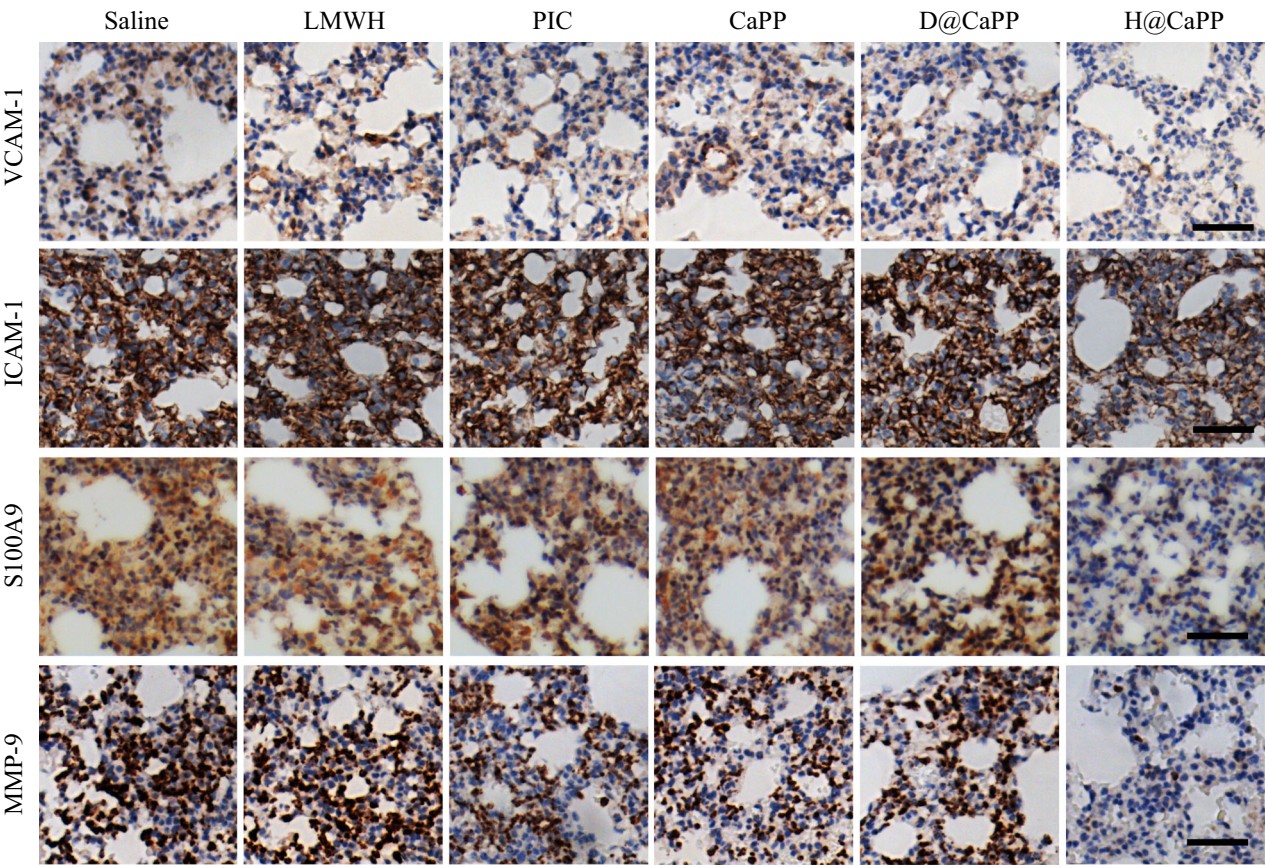

**Fig. 8 H@CaPP inhibited the formation of a pre-metastatic microenvironment in vivo.** Representative images of immunohistochemical analysis of ICAM-1, VCAM-1, MMP-9, and S100A9 (brown) in the lung of breast tumor-bearing mice after intravenously administrated with saline, LMWH, piceatannol, CaPP, D@CaPP, or H@CaPP (i.v. at a dose of piceatannol or PP at 0.020 mmol/kg, at a dose of Dextran or LMWH at 10 mg/kg) for seven times in 14 days. Scale bar, 50 μm.

administered with saline, LMWH, PIC, CaPP, D@CaPP, and H@CaPP, respectively. H&E staining analysis of major organs showed that after the treatment, the mice from the H@CaPP group did not show obvious irregularities or lesions (Supplementary Fig. 24a), indicating the low tissue-specific toxicity of H@CaPP. In addition, the safety property of the nanotherapeutic in combination with the two regular clinical treatments was further demonstrated by the similar bodyweight of mice from the various groups (Supplementary Fig. 24b, c). We also collected peripheral blood for hematological and serum biochemical analyses. As shown in Supplementary Fig. 25a–c, there were no significant downward trends of WBC (white blood cell) counts, RBC (red blood cell) counts and PLT (platelets) counts in the H@CaPP-treated group compared with the PBS group. This result suggested that the nanotherapeutic did not cause side effects on the hematological system. The biochemical analysis of the serum was also conducted to evaluate organ toxicity (Supplementary Fig. 25d–g). H@CaPP would not increase the levels of ALT (alanine aminotransferase), AST (aspartate trans-aminase), BUN (blood urea nitrogen), and CRE (creatinine), which indicated that the nanotherapeutic would not cause liver injury and kidney injury. Hence, the safety evaluation demonstrated that the nanotherapeutic was biocompatible and had clinical transformation potential.

**Pharmacokinetic study**. For verifying whether the nanotherapeutic could prolong the circulation time of the two agents, the pharmacokinetics of various formulations was evaluated in SD rats. Pharmacokinetics of LMWH and the formulation were evaluated by anti-FXa assay. The assay was utilized to estimate the concentration of LMWH in blood after administration. The obtained data were analyzed with the main pharmacokinetic parameters (including $AUC_{0-t}$, $k$, $t_{1/2}$, Cl) and the pharmacokinetic profiles were presented in Supplementary Table 3 and Supplementary Fig. 26, respectively. After loaded on the surface of the PEGylated nanoparticle (H@CaPP), the half-life blood circulation time of LWMH was obviously prolonged. Compared with LMWH ($t_{1/2}$ 2.71 h), H@CaPP exhibited a longer half-life blood circulation time ($t_{1/2}$ 4.09 h). In addition, LWMH coated on CaPP showed a slower clearance (Cl) rate and higher AUC (about 2.1 folds) when compared with free LMWH. The results demonstrated that H@CaPP exhibited a superior blood circulation profile compared with LMWH. To evaluate the pharmacokinetics of PIC/PP formulations in vivo, a high-performance liquid chromatography/electrospray ionization mass spectrometry method has been developed for the simultaneous determination of PP and piceatannol in rat plasma. As the results are shown in Supplementary Table 4 and Supplementary Fig. 27, the mean $AUC_{0-t}$(nmol/ml/h) of PIC, PP, CaPP, and H@CaPP were 2.64, 2.03, 7.51, and 16.59, respectively. And H@CaPP (7.82 h) exhibited obviously longer half-life blood circulation time compared with PIC (0.71 h), PP (0.48 h), and CaPP (3.71 h). These data collectively demonstrated that LMWH modification improved the blood circulation profile of CaPP, and the long-term blood circulation of nanotherapeutic would ensure better targeting effects to the whole process of metastasis.

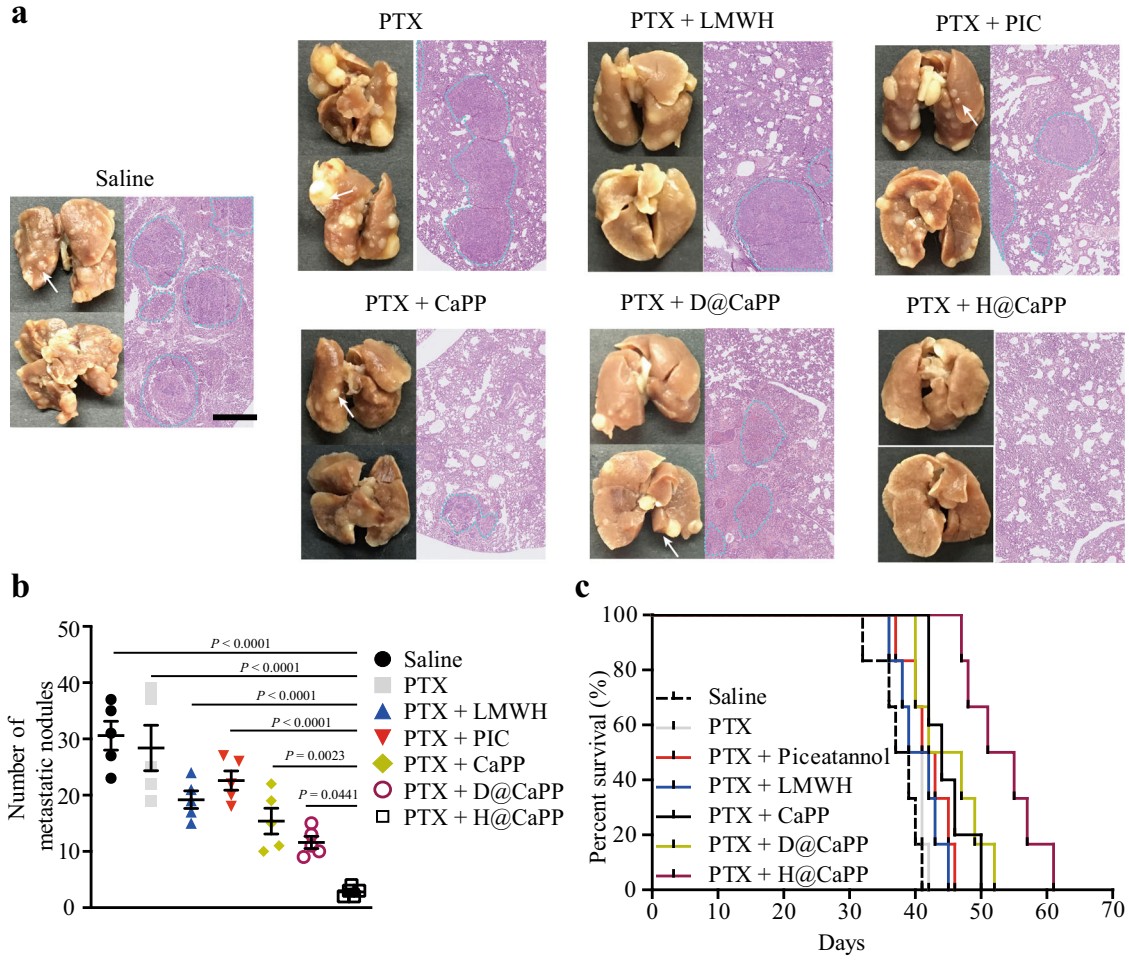

**Fig. 9 H@CaPP combined with chemotherapy achieved the anti-metastasis therapeutic effect in orthotropic primary tumor-bearing mice. a** Typical lung tissues with visualized metastatic nodules (white arrows) from each group, and representative fields of lung tissues with metastasis areas (blue dotted lines) from each group with H&E. Black scale bar, 5 mm; Scale bar, 0.5 mm. **b** The number of metastatic nodules from the mice after intravenously administration with saline, PTX, PTX + LMWH, PTX + PIC, PTX + CaPP, PTX + D@CaPP or PTX + H@CaPP (i.v. of PTX at $5.9 \times 10^{-3}$ mmol/kg, at a dose of piceatannol or PP at 0.020 mmol/kg, at a dose of Dextran or LMWH at 10 mg/kg) for seven times in 14 days (ANOVA, mean ± SD, $n = 5$ mice per group). **c** Survival curve of orthotropic breast tumor-bearing mice after intravenously administration with saline, PTX, PTX + LMWH, PTX + piceatannol, PTX + CaPP, PTX + D@CaPP, or PTX + H@CaPP for seven times during 14 days ($n = 6$ mice per group). One-way ANOVA with Tukey's multiple comparisons test (one-sided) was used for **b**. Error bars represent SD.

## Discussion

As most cancer patients die of tumor metastasis, how to effectively inhibit metastasis is critical to the clinical area. Unfortunately, according to clinical data, conventional clinical therapies, such as surgical resection and chemotherapy, fail to inhibit tumor metastasis[3,62–64]. The possible explanation behind this phenomenon could be that chemotherapy and resection cause an inflammatory response and promote the development of an immunosuppressive microenvironment via breaking the delicate balance between primary tumor and host physiology[4,5]. Thus, finding an effective anti-metastatic therapy reinforcing tumor surgical resection and chemotherapy is an urgent challenge in cancer therapy. For specifically combating the metastasis, recent studies focused on the analysis of the development of the metastasis[51,65]. For instance, monoclonal antibodies against the mesenchymal cadherin (N-cadherin) reduced invasion of cancer cells via hindering the EMT process[10]. And a dual-functionalized nanoparticle could inhibit the tumor neovasculature in the primary tumor and neutralize CTCs in the circulation[14]. The neutrophil-mimicking nanoparticles containing carfilzomib could hinder the colonization phase via inhibiting the progress of the pre-metastatic niche[13]. However, these strategies are so far compromised for their failure in hindering the whole process of metastasis including the invasion phase, circulation phase, and colonization phase. Thus, for systemically targeting and inhibiting metastasis, we combined the piceatannol and LMWH for synergistically hindering the multi-steps of metastasis[15,16,48,66]. Piceatannol was proved to be a potential pharmaceutical ingredient for some inflammation-related diseases, such as arrhythmia, atherosclerosis, and cardiovascular diseases. It could effectively hinder tumor cell invasion through impeding the EMT process (Fig. 5) and inhibited the adhesion of CTCs to activated ECs via reducing the expression of CAMs (VCAM-1 and ICAM-1) (Fig. 7). However, its clinical application is largely hindered by its poor targeting capability and quick elimination. LMWH, a complex mixture of glycosaminoglycans, is widely used for the prevention and management of acute venous thromboembolism (VTE). Aside from the effect, LMWH possesses anti-tumor potential via interference with tumor-induced hypercoagulation, tumor angiogenesis, and binding of platelets and tumor cells[32,67,68]. In addition, LMWH could bind and target to the ECs at the pre-metastatic sites with high expression of P-selectin (Figs. 4 and 6).

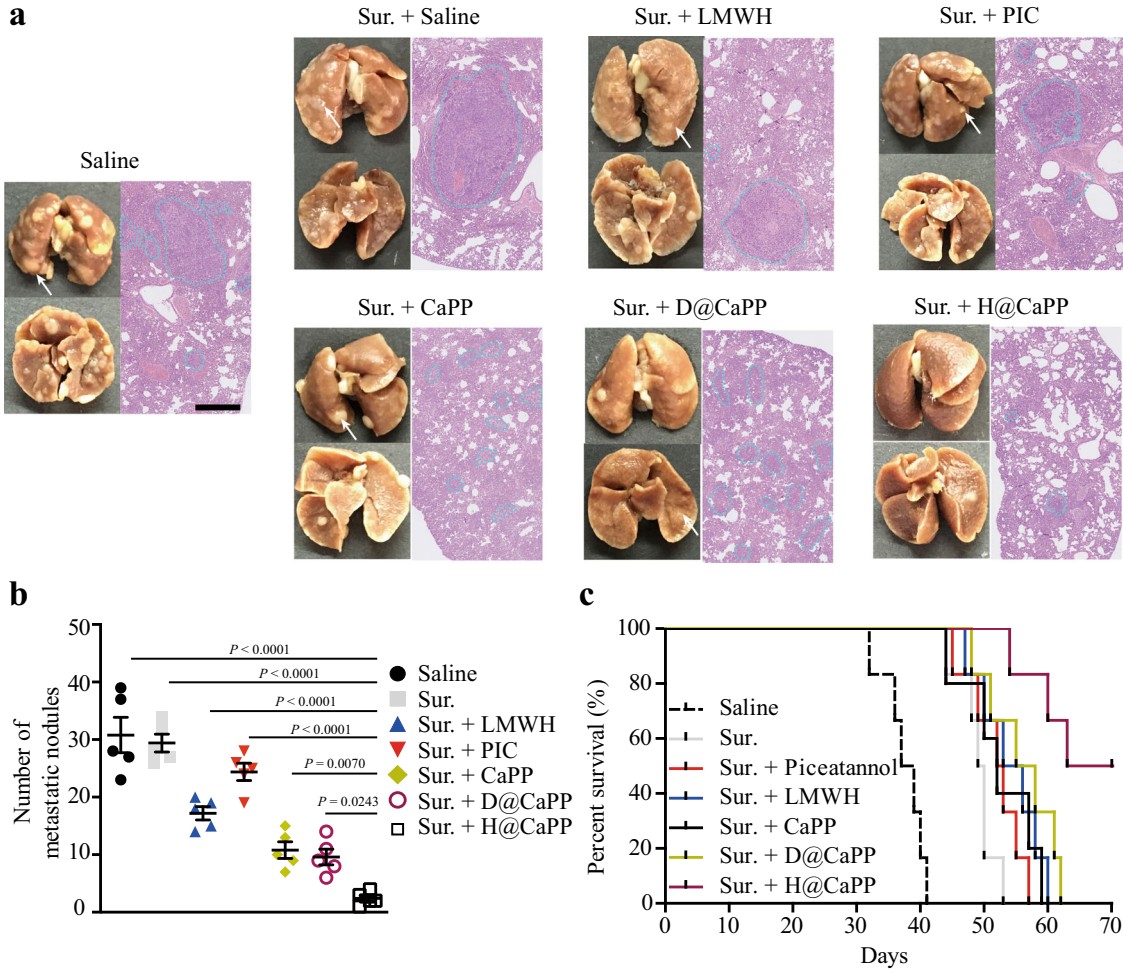

**Fig. 10 H@CaPP combined with surgery resection achieved the anti-metastasis therapeutic effect in orthotropic primary tumor-bearing mice. a** Typical lung tissues with visualized metastatic nodules (white arrows) from each group, and representative fields of lung tissues with metastasis areas (blue dotted lines) from each group with H&E. Black scale bar, 5 mm; Scale bar, 0.5 mm. **b** The number of metastatic nodules of the mice combined with surgical resection after intravenously administration with saline (without surgical resection), saline, LMWH, PIC, CaPP, D@CaPP or H@CaPP (i.v. at a dose of piceatannol or PP at 0.020 mmol/kg, at a dose of Dextran or LMWH at 10 mg/kg) for seven times during 14 days (ANOVA, mean ± SD, $n = 5$ mice per group). **c** Survival curve of orthotropic breast tumor-bearing mice combined surgery after intravenous administration with saline (without surgical resection), saline, LMWH, piceatannol, CaPP, D@CaPP or H@CaPP for seven times during 14 days ($n = 6$ mice per group). Error bars represent SD.

However, in vivo, LMWH could be quickly eliminated in circulation and induce acute coagulation disturbance after intravenous injection in vivo[24,69]. Hence, in order to hinder the whole process of metastasis, how to overcome the quick elimination and best integrate two agents into one targeting drug delivery system is a challenge.

Here, we developed a systemic metastasis-targeted nanoparticle which was loaded PP into lipid calcium core and further coated with LMWH via electrostatic interaction. PP could be released from lysosomes into the cytoplasm (Supplementary Fig. 12). Interestingly, it has been reported that cellular phosphatases are elevated in inflammation sites and tumor tissues[34,70]. Therefore, the conversion of PP to piceatannol could be enhanced following the co-delivery of H@CaPP at multiple steps of tumor metastasis[70]. The stability and the size of H@CaPP (Fig. 2) present adequate drug accumulation in the primary tumor through enhanced permeability and retention effect (EPR effect)[71]. LMWH, coated on the surface of H@CaPP, could not only hinder the circulation phase via the formation of "micro-thrombi" (Fig. 6) but also target the pre-metastatic niche in vivo (Fig. 4). Especially in vivo, H@CaPP achieved more inhibition of adhesion of platelets to tumor cells because it might enable a longer

circulating time of the LMWH (Fig. 6 and Supplementary Fig. 26, Supplementary Table 3). In addition, H@CaPP could effectively hinder the invasion phase via impeding the EMT process (Fig. 5) and hinder the colonization phase via reversing aberrant expression of pre-metastatic proteins in vivo (Fig. 8 and Supplementary Fig. 21). According to the results of the pharmacokinetic study (Supplementary Fig. 27 and Supplementary Table 4), PIC and PP were all cleared the fastest as expected. CaPP was cleared more quickly than H@CaPP probably because of its recognition by the mononuclear phagocyte system (MPS) due to the surface positive charge[72,73]. H@CaPP achieved greater inhibitory effects on the whole metastasis process that could be caused by the superiority in blood circulation profile and systemic targeting delivery.

To recapitulate the clinical conditions of tumor therapies, we selected orthotropic breast tumor-bearing mice with either chemotherapy or surgical resection as pre-clinical models to evaluate whether the H@CaPP has adjuvant potential. Not surprisingly, chemotherapy or surgical resection alone could not efficiently hinder the metastasis of TNBC. When combining with the nanoparticle, the treatment regimen exhibited a prominent effect on inhibiting tumor metastasis, and the efficacy of H@CaPP was

higher than that of non-targeting nanoparticles (D@CaPP) (Figs. 9 and 10) while limited toxicity to the major organs and the immune system was observed in H@CaPP in vivo (Supplementary Figs. 24 and 25). Collectively, combining H@CaPP with surgical resection or chemotherapy may be a safe and promising clinical strategy. The adjuvant nanotherapeutic might become a general solution for combating metastatic cancers, such as colon cancer, melanoma, and non-small cell lung cancer. Furthermore, the combination of adjuvant nanotherapeutic would provide a safe and metastasis-specific option in promoting and complementing other therapies, such as radiotherapy, immunotherapy, and stem cell therapy.

## Methods

**Reagents**. Dextran was purchased from Aladdin Co., Ltd (Shanghai, China). Low molecular weight heparin (LMWH) was obtained from Meilun Biotechnology Co., Ltd (Dalian, China). Hoechst 33258, toluidine blue, 1,1′-dioctadecyl-3,3,3′,3′-tetramethylindocarbocyanine perchlorate (DiI), 1,1′-dioctadecyl-3,3,30,3′-tetramethyl indotricarbocyanine iodide (DiR), and protease inhibitor cocktail were obtained from Sigma-Aldrich (St. Louis, MO, USA). D-luciferin potassium was bought from PerkinElmer Inc. (Waltham, MA, USA). Tumor necrosis factor-α (TNF-α) was purchased from Sinobio Co., Ltd (Shanghai, China). The phycoerythrin (PE) labeled anti-CD62P (P-selectin) antibody (1/20 dilution), PE-labeled anti-CD106 (VCAM-1) antibody (1/100 dilution), PE-labeled anti-CD62E (E-selectin) antibody (1/20 dilution) and PE-labeled anti-CD54 (ICAM-1) (1/20 dilution) antibody were provided by eBioscience. Anti-E-cadherin antibody (1/500,000 dilution for western blot and 1/80,000 dilution for immunohistochemistry), anti-vimentin antibody (1/1000 dilution for western blot and 1/250 dilution for immunohistochemistry), anti-GAPDH antibody (1/10,000 dilution), anti-CD62E (E-selectin) antibody (1/200 dilution), anti-CD62P (P-selectin) (1/500 dilution), anti-CD-106 (VCAM-1) antibody (1/500 dilution), anti-CD54 (ICAM-1) (1/2000 dilution), anti-CD41 antibody (1/500 dilution), anti-MMP-9 antibody (1/5000 dilution), anti-S100A9 antibody (1/500 dilution), anti-Ly6G antibody (1/2000 dilution), horseradish peroxidase-conjugated (HRP) secondary antibody (1/2000 dilution), and Alexa Flour 488-labeled secondary antibody (1/100 dilution) were bought from Abcam (Cambridge, MA, USA). Copper grids were purchased from Xinxing Braim Technology Co., Ltd (Beijing, China). Dulbecco's modified Eagle medium (DMEM), certified fetal bovine serum (FBS), penicillin-streptomycin stock solutions, and trypsin-EDTA (0.25%) were obtained from Invitrogen Co., Carlsbad, CA, USA. Calcein-AM was provided by Yeasen Biotech Co., Ltd (Shanghai, China). DSPE-PEG[2000] was provided by Yarebio Co., Ltd (Shanghai, China). Cholesterol (CHO), DOTAP, and dioleoyl phosphatidic acid (DOPA) were purchased from Avanti Polar Lipids (Alabaster, AL, USA). Piceatannol was provided by Biopurify Phytochemicals Co., Ltd (Chengdu, China). Paclitaxel (PTX) was provided by Jinhe Biotechnology Co., Ltd. (Shanghai, China). Bicinchoninic acid (BCA) protein assay kit and RIPA lysis buffer were provided by Beyotime Institute of Biotechnology (Jiangsu, China). Heparin LRT kit was provided by hyphen-BioMed Co., Ltd. (Neuville-sur-Oise, France). All the other chemical reagents and solvents were acquired from Sinopharm Chemical Reagent Co., Ltd (Shanghai, China).

**Cell lines and animals**. HUVECs, 4T1 cells, and 4T1-Luc[+] cells were obtained from the Chinese Academy of Sciences Cell Bank (Shanghai, China). The cells were maintained in DMEM medium, supplemented with 10% FBS (v/v), 100 mg/mL streptomycin, and 100 U/mL penicillin at 37 °C in a humidified atmosphere with 5% CO2.

Male SD rats (220–250 g, 2 months) and female BABL/c mice (20 ± 1 g, 6 weeks) were provided by BK Lab Animal Ltd. (Shanghai, China) and raised under standard housing condition (25 ± 1 °C, 50% relative humidity and 12 h/12 h dark/light cycle with access to food and water ad libitum). The animal experiments (Ethical approval number:2017-03-YJ-CJ-01) were performed in accordance with guidelines evaluated and approved by the Institutional Animal Care and Use Committee (IACUC), School of Pharmacy, Fudan University (Shanghai, China).

**Synthesis and characterization of piceatannol phosphate**. Piceatannol phosphate (PP) was synthesized with some modification. Briefly, 0.8 mmol piceatannol, DMAP, and Et3N were added at THF. Then, 23 mmol ClP(O)(OEt)2 was added to a dropwise ice water bath at 70 °C for 24 h. Next, the mixture was purified using column chromatography to obtain the ethyl-protected PP, which was characterized by [1]H NMR, [13]C HMR, and [31]P NMR.

Then, the 0.065 mmol ethyl-protected PP and trimethylsilyl bromide were mixed in dichloromethane. After 2 days of stirring, methanol was added and stirred for 1 h. Then, we use high preparative performance liquid chromatography to obtain PP, which was characterized by [1]H NMR, [13]C HMR, and [31]P NMR.

**Preparation of H@CaPP**. The water-in-oil microemulsion method was utilized to get CaPP core[36–39,42]. Briefly, adding 300 μL of 2.5 mM CaCl2 was dispersed into cyclohexane/igepal CO-520 (71:29, v/v) to generate the Ca phase. Similarly, for obtaining phosphate phase, 300 μL of 30 mg/mL PP and 20 mM DOPA were added into 20 mL cyclohexane/igepal CO-520 (71:29, v/v). After that, the two phases were mixed for 10 min at room temperature. Then, 500 μL of 20 mM DOPA was dispersed into 40 mL of the mixture. Subsequently, ethanol was added. Finally, the mixture above was submitted to centrifuge at 12,500 × g to collect the CaPP core.

To prepare the final CaPP, CaPP core was mixed with CHO, DOTAP, DSPE-PEG[2000]. The DiI or DiR -labeled CaPP or DiR-labeled CaPP was prepared by the same method with DiI or DiR added to the lipids.

To get H@CaPP, LMWH solution was added into the CaPP suspension, followed by stirring at room temperature[31]. Free LMWH was separated from H@CaPP with an ultrafiltration tube (MWCO 100 kDa, Millipore, USA). D@CaPP that was deserved as the negative control was prepared with the same method as H@CaPP.

**Characterization of H@CaPP**. The size and zeta potentials of CaPP, D@CaPP, and H@CaPP were determined by dynamic light scattering (DLS) using a Zeta Potential/Particle Sizer NICOMP 380 ZLS (Santa Barbara, California, USA) and the morphology was examined under a field emission transmission electron microscopy (Tecnai G2 F20 S-Twin).

To measure the encapsulation efficiency (EE) and loading capacity (LC) of H@CaPP, the samples were dissolved in lysis buffer, containing 2 mM EDTA and 0.1% Trixton-100 in pH 7.8 Tris. The EE and LC were calculated as follows:

$$EE(\%) = \frac{PP \; encapsulated \; in \; H@CaPP}{Total \; PP \; added} \times 100\%$$

$$LC(\%) = \frac{PP \; encapsulated \; in \; H@CaPP}{Weight \; of \; H@CaPP} \times 100\%$$

The amount of attached LMWH was determined by measuring the free LMWH using the toluidine blue assay[74–76]. The binding efficiency was calculated as follows:

$$Binding \; efficiency(\%) = \frac{LMWH_{total} - LMWH_{free}}{LMWH_{total}} \times 100\%$$

We evaluated the stability of nanotherapeutics in PBS and serum by the stability assay. D@CaPP or H@CaPP were suspended with 10% FBS or PBS for 24 h, respectively. At various time points (0, 2, 4, 8, 12, 24 h), the samples were collected to analyze the size and zeta potential changes of nanotherapeutics.

**Release study**. We evaluated the drug release of nanotherapeutics in PBS[45]. H@CaPP, D@CaPP, and PP were packaged in the dialysis bags (MWCO 13000 Da) and immersed in PBS (7.4) at 37 °C under gentle shaking. Then, for evaluating the profiles of PP release from H@CaPP at the intracellular conditions, H@CaPP were packaged in the dialysis bags and immersed in pH 5.0 or pH 6.5. At various time points (0, 2, 4, 8, 12, 24 h), the amount of PP released was determined by HPLC (Agilent 1200, USA). Similarly, we evaluated the LMWH release from the H@CaPP in PBS. The amount of LMWH was determined by the toluidine blue assay.

**Hemolysis study**. We performed the hemolytic assay to investigate the blood safety of H@CaPP[24,44]. The red blood cells were obtained from BALB/c mice. Samples incubated with 1% Triton X-100 served as the positive control and samples incubated with PBS as the negative control, respectively. Then, each sample was centrifuged at 800 × g for 15 min. The absorbance of the supernatant was analyzed at 540 nm by a microplate reader.

**PP conversion to piceatannol by 4T1 cells and HUVECs**. The conversion of PP to piceatannol was validated in 4T1 cells and HUVECs. H@CaPP containing 50 μg PP was added to 4T1 cells and HUVECs for 4 h, respectively. Then, the medium or cells were subjected to lyophilization in 1% Triton X-100. Then, piceatannol was detected by HPLC analysis.

**Invasion assay**. The invasion assay was performed using the Transwell system (Corning Costar Co., Cambridge, MA, USA). The upper side of the filter (8 mm pore size) was coated with Matrigel. The 4T1 cells (5 × 10[5]) were resuspended with samples in DMEM containing free piceatannol, LMWH, D@CaPP, and H@CaPP (at the concentration of 10 μM piceatannol/PP and 1.3 mg/mL LMWH) and placed in the upper part of the Transwell plate. After 24 h, the cells on the upper surface of the filter were completely wiped off, fixed with methanol for 15 min, and stained with 500 μL 0.5% crystal violet at 37 °C for 30 min.

**Adhesion assay**. HUVECs were seeded in 12-well plates (Cellvis, Mountain View, CA, 020012) with DMEM, respectively, containing TNF-α (50 ng/mL), TNF-α and LMWH (1.3 mg/mL), TNF-α and piceatannol (10 μM), TNF-α and D@CaPP (PP = 10 μM), TNF-α and H@CaPP (PP = 10 μM), for 24 h. Meanwhile, 4T1 cells were obtained and labeled with calcein-AM. Then, labeled-4T1 were added to the HUVECs. After incubating for 30 min at 37 °C with gentle shaking, the number of cells was quantified.

4T1 cells were seeded in 12-well plates with DMEM respectively containing piceatannol (10 μM), LMWH (1.3 mg/mL), D@CaPP (piceatannol = 10 μM) and

H@CaPP (LMWH = 1.3 mg/mL, piceatannol = 10 μM) for 24 h. Meanwhile, platelet-rich plasma was isolated through gradient centrifugation from the whole blood of female BALB/c mice. Then, platelets were obtained and labeled with calcein-AM. Then, the platelets were added to the 4T1 cells staining nuclei by Hoechst 33258 (blue). After incubating for 30 min at 37 °C with gentle shaking, the fluorescent intensity of platelets was quantified.

**The construction of the pre-metastatic model in vivo**. We established the orthotropic breast tumor-bearing mice by inoculating 4T1 cells into the mammary fat pat. Tumor-bearing mice were sacrificed on days 2, 4, 6, 8, 12, 10, 14, and 16 after tumor cell inoculation. Then, obtained lung sections of tumor-bearing mice were served for hematoxylin and eosin (H&E) staining. And the lung sections were served for anti-MMP-9 antibody, anti-S100A9 antibody, and HRP double staining.

**Biodistribution of H@CaPP in breast tumor-bearing mice**. Briefly, $3 \times 10^6$ 4T1 cells were inoculated into one of the mammary fat pads of BALB/c mice. The mice were intravenously injected with free DiR, D@CaPP, or H@CaPP at the dose of 1 mg/kg DiR, respectively. The biodistribution of nanoparticles in tumor-bearing mice was detected at predetermined time points (2, 4, 8, 12, and 24 h). Then, the major organs (heart, liver, spleen, lung, and kidney) and tumor tissues of the mice were harvested for ex vivo imaging.

**Pre-metastatic region formulating and homing assay**. To further reveal the distribution of nanotherapeutics in the pre-metastatic site, DiI-labeled D@CaPP and H@CaPP were intravenously given to orthotropic tumor-bearing mice and normal mice. Twenty-four hours later, the lungs were resected, fixed by 4% paraformaldehyde for 48 h, embedded by O.C.T, frozen in −80 °C for 12 h, and sliced. The sections were incubated with anti-E-selectin antibody and P-selectin antibody in 4 °C for 12 h. The sections were stained with an Alexa 488-labeled secondary antibody (green) for 2 h at room temperature and Hoechst 33258 that was used to stain the cell nucleus (blue) for 8 min at room temperature before the subjection to confocal microscopy.

**The vivo therapeutic effect in combination with the surgery model**. We conducted a preliminary experiment to determine the optimal time of surgery in breast cancer-bearing mice and the impact of resection on the survival time of the mice. 3 × 10^6 4T1-Luc^+ cells were inoculated into one of the mammary fat pads of 20 BALB/c mice[14]. The control (first) group did not undergo the surgery after inoculation, the second group underwent the surgery on the 8th day after inoculation, the third group underwent the surgery on the 11th day after inoculation, and the fourth group underwent the surgery on the 14th day after inoculation[77]. The procedure of surgery resection was as below: (i) An incision was cut on the tumor skin. (ii) The tumor was removed. (iii) Tumor was cut off with scalpel from the underlying tissue. (iv) The surgery wound was enclosed with sutures. For avoiding the death of mice caused by the operation, tumor removal surgery was carried out under the guidance of a professional surgeon. After that, the mice from all groups were sacrificed with lungs on 35th day after inoculation. The metastasis nodules in the lungs were recorded.

We further turn to the anti-metastatic efficacy of nanotherapeutics in orthotropic breast tumor resected mice. The seven groups of mice were intravenously administrated with saline (two groups), piceatannol, LMWH, CaPP, D@CaPP, or H@CaPP (at a dose of piceatannol or PP at 0.020 mmol/kg, at a dose of Dextran or LMWH at 10 mg/kg) for seven times during 14 days, respectively. Except for one of the saline groups setting as a non-surgery control group, other groups underwent the surgery of tumor resection on the 8th day after inoculation. Sixteen days after the last administration, the lungs of the mice ($n = 5$) were excised and embedded in paraffin. The obtained sections were served for hematoxylin and eosin (H&E) staining. The six tumor-bearing mice were monitored continuously and the survival time was recorded.

**The vivo therapeutic effect in combination with the chemotherapy model**. We also explored the anti-metastatic efficacy of nanotherapeutic combined with chemotherapy in breast tumor-bearing mice. The mice were randomly divided into seven groups. When the size of the tumor reached approximate 30 mm^3, the seven groups of mice were administrated with saline, PTX, PTX and piceatannol, PTX and LMWH, PTX, and CaPP, PTX and D@CaPP, or PTX and H@CaPP (PTX at $5.9 \times 10^{-3}$ mmol/kg, at a dose of piceatannol or PP at 0.020 mmol/kg, at a dose of Dextran or LMWH at 10 mg/kg) for seven times during 14 days, respectively. Sixteen days after the last administration, the lung sections of mice were served for H&E staining. The six tumor-bearing mice were monitored and the survival time was recorded.

**Immunohistochemistry for tumor and lung sections**. 4T1 cells ($3 \times 10^6$) were carefully inoculated into one of the mammary fat pads of BALB/c mice. The mice were randomly divided into five groups. When the size of tumors reached to approximate 30 mm^3, the six groups of mice were intravenously administrated with saline, piceatannol, LMWH, CaPP, D@CaPP, or H@CaPP (at a dose of piceatannol or PP at 0.020 mmol/kg, at a dose of Dextran or LMWH at 10 mg/kg) for seven

times during 14 days, respectively. After the administration, the obtained sections of tumors were served for anti-CD41 antibody, anti-E-cadherin antibody, anti-vimentin antibody, and HRP double staining. The obtained sections of lungs were saved for anti-MMP-9 antibody, anti-ICAM-1 antibody, anti-VCAM-1 antibody, anti-S100A9 antibody, anti-Ly6G antibody, and HRP double staining.

**Inhibition of EMT process of tumor cells by western blot in vitro**. 4T1 cells were seeded in 6-well plates. After 12 h, 4T1 cells were incubated with DMEM containing PBS, LMWH, piceatannol, D@CaPP, or H@CaPP (dose of free piceatannol or PP at 10 μM, dose of Dextran or LMWH at 1.3 mg/mL) for 24 h. Equal proteins from the groups were electrophoresed at the 8% sodium dodecyl sulfate (SDS)-polyacrylamide Bis-Tris gel at 100 V, and then transferred to nitrocellulose membranes at 290 mA. The membranes were incubated at 4 °C with anti-Vimentin antibody, anti-E-Cadherin antibody, and anti-GAPDH antibody for 12 h. Then, the nitrocellulose membranes were incubated with the horseradish peroxidase (HRP)-conjugated secondary antibody. After washing with TBST, the labeled proteins were visualized by ECL kits. The western blot images were acquired via Image Lab 6.0.1.

**In vitro cellular uptake**. HUVECs were seeded in 96-well plates. The cells were stimulated by TNF-α for 4 h. After that, HUVECs were incubated with DiI-labeled D@CaPP, DiI-labeled H@CaPP, DiI-labeled D@CaPP + P-selectin antibody, DiI-labeled H@CaPP + P-selectin antibody, DiI-labeled D@CaPP + E-selectin antibody, and DiI-labeled H@CaPP + E-selectin antibody for 2 h. And then, the cells were washed twice with PBS (pH 7.4) and stained with Hoechst 33258 for 8 min[43].

**Flow cytometry assay for expression of adhesion molecule in vitro**. HUVECs were resuspended with DMEM containing TNF-α, TNF-α and piceatannol, TNF-α and LMWH, TNF-α and D@CaPP, TNF-α and H@CaPP (TNF-α = 50 ng/mL, a dose of free piceatannol or PP at 10 μM, dose of Dextran or LMWH at 1.3 mg/mL) and inoculated into 24-well plates, respectively. The cells were incubated at 37 °C for 24 h. Then, the cells were trypsinized from the well plate and washed with PBS twice. After that, the cells were incubated with PE-labeled flow direct antibody against ICAM-1 and VCAM-1. The cells were washed three times and collected for analysis by flow cytometer (CytoFlex S, Beckman). And the results were analyzed via FlowJo (version 10.4) software.

**Toxicity evaluation**. The toxicity of the nanotherapeutics was investigated. The mice were administrated with saline, piceatannol, LMWH, CaPP, D@CaPP, or H@CaPP (at a dose of piceatannol or PP at 0.020 mmol/kg, at a dose of Dextran or LMWH at 10 mg/kg), respectively, seven times within 14 days as described above. The sections of their organs were followed by H&E staining. Both whole blood and serum were collected in EDTA-2K anticoagulative tubes. Whole blood cellular components were counted and compared (Mindray, BC-2800Vet, China). Serum AST, ALT, BUN, and CRE in the serum were assayed (Rayto, Chemray 240, China).

**Pharmacokinetic study**. In order to investigate the pharmacokinetics of LMWH in vivo, six SD rats were randomly assigned to 2 groups and intravenously administrated with LMWH and H@CaPP, respectively, at a dose of 100 IU (anti-FXa)/kg. About 0.5 mL blood samples were drawn from femoral artery cannula at 0.083, 0.25, 0.5, 1, 2, 4, 8, 12, and 24 h after injection. The blood samples were collected in 1.5 mL 10% sodium citrate tubes and immediately centrifuged at 500 × g for 15 min, and then the obtained plasma samples were stored at −80 °C until analysis. The concentration of LMWH in rat plasma was analyzed via anti-FXa chromogenic assay by Heparin LRT kit[78,79]. For investigating the pharmacokinetics of piceatannol or piceatannol phosphate in vivo, 12 SD rats were randomly assigned to 4 groups and intravenously administrated with piceatannol, piceatannol phosphate, CaPP and H@CaPP, respectively, at a dose of 0.02 mmol/kg. About 0.4 mL blood samples were drawn from femoral artery cannula at 0.083, 0.25, 0.5, 1, 2, 4, 8, 12, 24 h after injection. The blood samples were collected in 1.5 mL heparinized polythene tubes and immediately centrifuged at 2200 × g for 8 min, and then the obtained plasma samples were stored at −20 °C until analysis. High-performance liquid chromatography/electrospray ionization mass spectrometry method has been developed for the simultaneous determination of PP and Piceatannol in rat plasma. The protein precipitation method was used to extract PP, Piceatannol, and internal standard from rat plasma. The Triple Quad™ 6500 triple-quadrupole tandem mass spectrometry (SCIEX, USA) was operated under the multiple reaction-monitoring mode (MRM) using the negative electrospray ionization technique. Detection of the ions has monitored the transition of the $m/z$ 304.5 precursor ion to the $m/z$ 130.8 for PP, $m/z$ 243.0 precursor ion to the $m/z$ 201.0 for piceatannol, and $m/z$ 307.2 precursor ion to the $m/z$ 161.1 for internal standard warfarin. Chromatographic separation was achieved on a Venusil XBP Phenyl column (100 mm × 2.1 mm, 5 μm, Agela Technologies) by using gradient elution with a mobile phase consisting of 0.1% formic acid in water (mobile phase A) and 0.1% formic acid in methanol (mobile phase B). Data processing was achieved on Analyst® software, version 1.6.3 (SCIEX, USA). The pharmacokinetic parameters were analyzed by model-independent method and calculated by Microsoft Excel (version 16.45) and PKSolver (version 2.0).

**Statistical analysis**. All statistical analysis was performed using GraphPad Prism 8.4.0 software. All results were presented as the mean ± standard deviation (SD). The Student's *t*-test was used for a two-group comparison. The difference among multiple groups was determined by one-way ANOVA analysis followed by Tukey's multiple comparisons test.

**Reporting summary**. Further information on research design is available in the Nature Research Reporting Summary linked to this article.

## Data availability

All the other data supporting the findings of this study are available within the article and its Supplementary Information files and from the corresponding author upon reasonable request. A reporting summary for this article is available as a Supplementary Information file. Source data are provided with this paper.

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

## Acknowledgements

This work was supported by the National Natural Science Foundation of China (Nos. 92068110, 81690263, 81673019, 81973272, 81773909), grant from Shanghai Science and Technology Committee (20JC1411800, 19410710100), Shanghai talent development funds (201665), Excellent Medical Profession Scholarship of Shanghai municipal commission of health and family planning (2017YQ060).

## Author contributions

J.C., X.G., and M.X. designed the research; M.X. carried out most experiments; J.C., X.G., K.H., Y.H., and S.L. provided advice for writing the article; Y.L., N.M., and H.L. provided help for editing the manuscript; J.C., M.X., Y.H., and Y.C. analyzed the data; Y.C. and S.Z. assisted with the establishment of orthotropic breast tumor-bearing mice model; Y.L. provided help for scientific drawings; D.W. and Q.Z. assisted with the tumor frozen microtome; F.L. assisted with the pharmacokinetic study; M.X., X.G., and J.C. wrote the paper. All authors discussed the results and commented on the manuscript. Both J.C. (orcid.org/0000-0003-1330-9616) and X.G. (orcid.org/0000-0001-6789-3059) are corresponding authors.

## Competing interests
The authors declare no competing interests.
