## [Peer Review File · Nature Communications]

REVIEWER COMMENTS

Reviewer #1 (Remarks to the Author):

Tumor metastasis is one major reason for the failure of clinical tumor therapy. In this study, Xu and co-workers reported a systemic metastasis-targeted nanotherapeutic (H@CaPP) for hindering tumor metastasis.

I like this study in the following three points:

(1) The structure of the nanotherapeutic (H@CaPP) is quite simple, with phosphorylated piceatannol and calcium as the inner core, while low molecular weight heparin decorated on the surface via electrostatic interaction.

(2) The authors provided abundant evidence to prove that the nanotherapeutic is effective to all the three key steps of tumor metastasis: the invasion phase, the circulation phase and the colonization phase.

(3) The authors carefully evaluated a spontaneous tumor metastasis model in mice (they examined the lung tissue every two days from primary tumor establishment, Fig.S11a), and therefore defined the pre-metastatic stage and later stages. This provides reference value to tumor metastasis study in mouse models.

The authors did many mechanism studies in this manuscript. The conclusions are mostly supported by these data. The manuscript is well-organized and well-written.

I have the following concerns for this manuscript:

1. The structure of the H@CaPP nanoparticles need to be further clarified. If DSPE-PEG2k was used in preparing the CaPP, then PEG should be on the outside after hydration. The scheme and colors used in Fig1a is confusing. In addition, if there is PEG on the CaPP surface, the absorption efficiency and the stability of LMWH on the CaPP may be affected. The heprin is at the outer space of PEG or embeded in PEG? Please clarify on this structure and provide quantitative contents of these components.

2. In Fig1d, TEM images showe a dimmer layer on the surface of H@CaPP and there is a gap between the layer and the core. Please give more explanations on this structure. Why these is dimmer layer? Why there is a gap between the layer and the core?

3. In Fig1f in vitro release study, only 25% percent PP released from the nanoparticles in 24 h in PBS (pH 7.4). Please provide more release results at conditions mimicking the intracellular condition.

4. How about the stability of heprin on the H@CaPP? A release study of heprin may help to explain.

5. Using heparin to inhibit the formation of “micro-thrombi” in circulation is interesting. But since platelets are abundant in blood, how much heparin in H@CaPP could act on the “micro-thrombi”? Will this interaction between heparin and platelets lead to off-target effect and lead the nanotherapeutic to other places?
6. Fig5a is an interesting in vitro study to prove the inhibitory effect of H@CaPP on the adhesion of platelets to tumor cells. Could the “micro-thrombi” be isolated or identified from the blood? Because the in vitro study is still not enough to prove the H@CaPP acts on the “circulation phase” but not later phases.
7. Is Fig5c the primary tumor or metastatic tumor? The Figure shown in the H@CaPP group is strange since tumor as an inflammatory tissue is known for platelet aggregation and thrombi formation.
8. Blood safety test (for example, hemolysis study) should be provided for this nanotherapeutic.
9. The title of this manuscript may be re-considered, since the emphasis of this study is using a nanotherapeutic to target the three stages of metastasis-but not reinforce surgery and chemotherapy.
10. Page 5 line 135, there is no Scheme 1b in Scheme 1.
11. Fig S11a is interesting. Please clarify the n numbers in each group and time point. Is it reproducible and even in different mice?
12. Page 10 line 243, I think the citation for Fig3d should Fig3c.
13. Page 17, line 388, “. And H@CaPP” is mistake.

Reviewer #2 (Remarks to the Author):

In this manuscript, Xu and co-authors designed and developed a systemic metastasis-targeted nanotherapeutic for delivering a combination of anti-inflammatory and anti-coagulant drugs to disrupt the process of tumor metastasis. They claimed that this effectively hinders tumor metastasis via blocking the EMT process during tumor invasion phase and decreasing the formation of micro-thrombi during tumor colonization phase. They also showed that, when combined with other cancer treatment such as chemotherapy or surgical resection of tumor, the nanotherapeutic significantly reduced cancer metastasis and improved survival time. The work is comprehensive. However, this reviewer has some major concerns on the targeting strategy, the release of the active pharmaceutical ingredient (API), the anti-micro-thrombi effect of heparin, and the anti-inflammatory effect of piceatannol together with its effect on E-cadherin, vimentin, vcam-1 etc. These together with other concerns are detailed below.

1. The authors claimed the ant-inflammatory effect is an important mechanism of their approach. However, no data on anti-inflammation of the H@CaPP nanoparticles is shown. Data on how the nanoparticle affects the hallmarks of inflammation (neutrophil and monocyte and their recruitment), should be provided. The mechanism of piceatannol on E-Cadherin, Vimentin, Vcam-1 and ICAM-1 are not provided or described clearly in this study either. The authors should provide more information on how piceatannol can regulate E-cadherin, vimentin, vcam-1 and ICAM-1 expression in 4T1 cells and ECs.
2. In Figure 5, they show the LMWH on the H@CaPP nanoparticles could inhibit the adhesion of platelets. Since platelets are abundant in the blood, how can the nanoparticles escape from binding to them in blood before reaching tumor? After the nanoparticles reach tumor and are taken up by activated ECs, how can they be effective in cancer cells and still regulate the expression of vimentin and E-cadherin?
3. The general knowledge is that heparin only makes blood thinner and decreases the clotting ability of blood as is also shown by the authors in Fig. 5, but it does not de-clot micro-thrombi already formed in blood. Since the authors claimed the anti-micro-thrombi ability of their H@CaPP nanoparticles, this should be demonstrated in vitro.
4. The criteria for identifying pre-metastatic, invasion, micro-thrombi, and colonization phase are unclear at best. How can the authors specify the metastatic phase and do the treatment?
5. The authors assumed that by targeting multiple steps of the EMT process, they can effectively reduce/inhibit cancer metastasis. In a recent study by Fischer et al (doi: 10.1038/nature15748), it is shown that EMT inhibition in lung cancer did not impact lung metastasis, and instead, it contributes to the chemoresistance. This contradicts the authors' claim of targeting the EMT for reducing lung metastasis. Although the authors show the data of the H@CaPP accumulation into the lungs and tumor, they do not show any data to prove that once the EMT process is hindered, the metastatic tumor cells become non-metastatic.
6. To maintain the multifunctional efficacy in different tumor phase and different cells, pharmacokinetics of LMWH and piceatannol from the H@CaPP nanoparticles should be provided. Also, the in vitro release profile at tumor pH should be included.
7. In the discussion part, the authors mentioned that PP could be dephosphorylated by phosphatases that are extensively expressed by cancer cells. How was the expression of phosphatases in activated ECs and 4T1 cells?
8. The schematic illustration is unclear. More explanation is needed in the caption to explain what is going on in the schematic including the three targeting phases.
9. The authors should explain more on the low radiant efficiency of the H@CaPP in the tumor/area in Fig. 2d. The image in Fig. 2c clearly shows that the ex vivo fluorescence of the tumor is higher than the rest according to the intensity bar. There is also a confusion in Fig. 4b, where the western blot data does not match the caption.
10. In Fig. 7, there is only a minor exhibition of VCAM-1 inhibition by H@CaPP and an insignificant inhibition of ICAM-1. Please clarify.

11. The authors should review carefully and fix many grammatical errors such as those in (391-392), (410), 439, (449-450), Fig.2d (kidney?), Fig S14 (92); they should correct the caption in line 399-400 (Left photo is fluorescence image, not bright field as they stated) and proofread the manuscript carefully. The authors also mentioned the use of AVONA in (Fig. 4, 313 &317) for their statistical analysis. I believe it should be ANOVA. Number of mice per group for all group (e.g. survival experiments) should be clarified. Also, the manuscript would benefit a lot from a careful proofreading by a native English speaker.

Reviewer #3 (Remarks to the Author):

Minjun Xu and colleagues have submitted an article in which they present a new systemic metastasis-targeted nanotherapeutic that may reinforce tumor surgical resection and chemotherapy. This new treatment may target various steps of metastases development and implantation, suggesting an overall activity in the metastatic process, in comparison to other current available therapeutic approaches.

This study presents noteworthy results due to the fact that the new developed therapeutic approach possesses specific molecular activities at different levels of the metastatic process, suggesting a higher efficacy, as well as a higher potential at different steps of cancer patient's clinical history. Hence, such an observation is of significance to the field; however, various other studies have already been published, in which different mechanisms of action have been identified in the efficacy of new anti-metastatic therapies, particularly using nanoparticles, some of them being referenced by the authors, and some others, not (for instance H. Sun et al, *Int J Biol Macromol* 2018 ; Guo O et al, *J Biomed Nanotechnol* 2014). It has to be mentioned that almost all of those publications were based on a sole preclinical model using 4T1 breast carcinoma cells, that one used in the study of Xu and colleagues.

The methodology of the paper is sound, and the work clearly supports the conclusions. Moreover, all experiments have been performed in accordance with all ethical requirements; one point requests a specific mention by the authors: in the figure S14, the authors have validated the surgery resection of the tumors in non-sacrificed mice to allow follow-up of metastases development. Hence, because of the high amount of deaths occurring during this experiment, this specific optimization of the in vivo protocol might have been previously approved by an ethics committee, and this approval should be added in the text.

I won't address chemical issues and specific validations which are not in my skeels. In contrast, I have few questions and requirements that may improve the article:

- In the Figure 2, there is a discrepancy between Figures c and d regarding fluorescence of D@CaPP and H@CaPP in the kidney.

- Looking at IHC slides of the Figure 4F, it is not clear how real difference could be noted between CaPP and H@CaPP. The authors may therefore explain which quantification has been performed? A similar issue on quantification is raised for the Figure 7.

- Concerning in vivo survival data, it is not clear how the authors have exactly processed. In the M&M section, it is mentioned that « the 60-days survival period of each group... ; line 753) has been checked. It might be explain, in such a situation, how the authors were able to follow mice survival and concomitantly obtain lung materials available for metastatic assessment? Indeed, they should have been confronted to the issue that some mice dead before any lung obtention. The paper did not mention the number of mice per group, but we can define it between 5 or 6. But, how many lungs have been studied per group? We have not the infotmation. Yet, this point is important because it is well known that there is heterogeneity between mice for which a sufficient number of animal is required for any molecular study (at least 3 mice).

- In the in vivo studies (Figures 8 and 9), histopathological analyses only concern lung and the occurrence of metastases. It might be useful to add tumor study (in terms of EMT) and lungs (in terms of metastatic niches as well as various markers previously studied in the paper, i.e. E-selectin, P-selectin, CD41, VCAM-1, ICAM-1, S100A9, and MMP9). Those results may definitively confirmed the anti-metastatic activity of H@CaPP.

- Finally, in the toxicity assessment, it lacks determination of hepatic, haematological and renal data. Moreover, the Figure 10 might be moved to the Supplementary section.

Point-to-point responses to reviewer's comments

Reviewer #1

1. The structure of the H@CaPP nanoparticles need to be further clarified. If DSPE-PEG2k was used in preparing the CaPP, then PEG should be on the outside after hydration. The scheme and colors used in Fig1a is confusing. In addition, if there is PEG on the CaPP surface, the absorption efficiency and the stability of LMWH on the CaPP may be affected. The heprin is at the outer space of PEG or embeded in PEG? Please clarify on this structure and provide quantitative contents of these components.

Response: Thanks for your comments and questions. Accordingly, we have revised the scheme and colors used in the revised Figure 1a. As mentioned by the reviewer, DSPE-PEG2k was used in preparing the calcium-phosphate liposome (CaPP) and located on the surface of CaPP to maintain the morphological stability of the nanoparticle and prolong the circulation time of the active pharmaceutical ingredient^{1, 2, 3}. LMWH was believed to be absorbed on the outer space of PEG through the electrostatic interaction between LMWH and CaPP. As shown in the revised Figure 1b-d, the particle size of H@CaPP was obviously larger than that of CaPP and the LMWH layer (around 15 nm) was observed on the surface of the nanoparticle. And the zeta potential of CaPP is approximately 25 mV (Figure 1e), while that of H@CaPP is approximately -37 mV (Figure 1e). The results suggested that PEG2k on the outside of CaPP would not hinder the electrostatic interaction between LMWH and CaPP. In addition, according to the Figure 1h and Figure S9, H@CaPP exhibited negligible changes in zeta potential and less than 16% of LMWH released from H@CaPP following 24h incubation in PBS, indicating that the heparin on H@CaPP was rather stable. For quantitative analysis of these components, we measured the binding efficiency (%) of LMWH by the toluidine blue assay^{4, 5, 6} and the binding efficiency of LMWH was $68.24\% \pm 5.04\%$. In addition, in H@CaPP, PEG accounted for around 29% (w/w) and LMWH accounted for around 17% (w/w) of the nanoparticle. Related description please see Line 164-190 (Page 7) in the revised manuscript.

Figure 1. Preparation and characterization of H@CaPP. **a** Schematic illustration of the preparation route of H@CaPP. **b** Particle size of CaPP, D@CaPP and H@CaPP measured by dynamic light scattering. **c** Average diameter of CaPP, D@CaPP and H@CaPP. **d** Morphology of CaPP and H@CaPP. Scale bar, 50 nm. **e** Surface zeta potential of CaPP, D@CaPP and H@CaPP. **f** The PP release profile of D@CaPP, H@CaPP or PP in PBS (pH 7.4) at 37°C with gentle stirring. **g** Average size of

D@CaPP and H@CaPP within 24 h incubation with 10% FBS or PBS (pH 7.4) at 37°C, respectively. **h** Zeta potential of D@CaPP and H@CaPP incubated with 10% FBS or PBS (pH 7.4) at 37°C, respectively within 24 h (mean \pm SD, n = 3).

Figure S9. The release of LMWH from H@CaPP in PBS within 24 h (mean \pm SD, n = 3).

2. In Fig1d, TEM images showed a dimmer layer on the surface of H@CaPP and there is a gap between the layer and the core. Please give more explanations on this structure. Why these is dimmer layer? Why there is a gap between the layer and the core?

Response: Thanks for your questions and suggestions. We speculate that the core is CaPP and the dimmer layer is LMWH. Previous work found that nanostructure prepared through the similar electrostatic interaction technique showed the dimmer layer on the surface^{7,8}. Here, CaPP showed positive zeta potential (Figure 1e) and the LMWH was adsorbed on the surface of CaPP via electrostatic interaction to obtain H@CaPP. The layer of LMWH is dimmer because its structure is looser than CaPP core. The gap might be the PEG chain. The one reason is there is only PEG between the lipid layer and LMWH. The other reason is that the CaPP core and LMWH could not be stained by the electron-dense stain (phosphotungstic acid) via negative staining technique and their colors were white or gray at the TEM images^{9,10}. While the PEG chain could be stained by phosphotungstic acid due to its low density, the color of PEG was black at the TEM images. And as shown in the Figure 1e, the zeta potential of CaPP is approximately 25 mV (Figure 1e). Then, after the negatively charged LMWH coated on CaPP via electrostatic interaction, the zeta potential of H@CaPP is approximately -37 mV (Figure 1e). The results showed that PEG2k on the outside of CaPP would not hinder the electrostatic interaction between LMWH and CaPP. Related results and analyses please see Line 164-173 (Page 7) in the revised manuscript.

Figure 1. Preparation and characterization of H@CaPP. **a** Schematic illustration of the preparation route of H@CaPP. **b** Particle size of CaPP, D@CaPP and H@CaPP measured by dynamic light scattering. **c** Average diameter of CaPP, D@CaPP and H@CaPP. **d** Morphology of CaPP and H@CaPP. Scale bar, 50 nm. **e** Surface zeta potential of CaPP, D@CaPP and H@CaPP. **f** The PP release profile of D@CaPP, H@CaPP or PP in PBS (pH 7.4) at 37°C with gentle stirring. **g** Average size of

D@CaPP and H@CaPP within 24 h incubation with 10% FBS or PBS (pH 7.4) at 37°C, respectively. **h** Zeta potential of D@CaPP and H@CaPP incubated with 10% FBS or PBS (pH 7.4) at 37°C, respectively within 24 h (mean ± SD, n = 3).

3. In Fig1f in vitro release study, only 25% percent PP released from the nanoparticles in 24 h in PBS (pH 7.4). Please provide more release results at conditions mimicking the intracellular condition.

Response: Thank you for the suggestion. Accordingly, we have performed additional experiments to determine the profiles of PP release from H@CaPP at the intracellular conditions. As reported, the calcium-phosphate liposome would be internalized by endosomal/lysosomal pathway^{11, 12, 13}. Therefore, we evaluated the PP release from H@CaPP in phosphate buffer solution at pH 6.5 (mimicking endosomes) and 5.0 (mimicking lysosomes). H@CaPP was packaged in the dialysis bags (MWCO 13000 Da) under gentle shaking. At various time points (0, 2, 4, 8, 12, 24h), the amount of PP released was determined. In pH of lysosomes (5.0), the H@CaPP released 83.0% of PP in 24 h. Whereas, in pH of lysosomes (6.5), only 35.5% of PP was released from the H@CaPP in 24 h. The result (Figure S10) indicated that the release rate of PP was significantly dependent on the pH value and the drug was released from lysosomes into cytoplasm. Related description please see Line 182-190 (Page 7) in the revised manuscript and the method has now been incorporated in the revised manuscript please see Line 731-736 (Page 34).

Figure S10. The PP release profile of H@CaPP in different pH medium (5.0 or 6.5) within 24 h (mean ± SD, n = 3).

4. How about the stability of heprin on the H@CaPP? A release study of heprin may help to explain.

Response: Thanks for the question and suggestion. In order to study the stability of the surface-coated heparin, we further determined the release of LMWH from H@CaPP in PBS (mimicking normal condition). H@CaPP and LMWH were packaged in the dialysis bags (MWCO 13000 Da) and immersed in PBS at 37°C

under gentle shaking. At various time points (0, 2, 4, 8, 12, 24h), the amount of LMWH released was determined by the toluidine blue assay. As shown in Figure S9, LMWH showed a burst release in the first 2 h and reached a plateau at 4 h. In contrast, less than 16% of LMWH was released from H@CaPP after 24 h incubation. According to the results, the heparin on H@CaPP achieved great stability in the circulation. Related description please see Line 177-182 (Page 7) in the revised manuscript and the method has now been incorporated in the revised manuscript please see Line 731-738 (Page 34).

Figure S9. The LMWH release from H@CaPP in PBS within 24 h (mean \pm SD, n = 3).

5. Using heparin to inhibit the formation of “micro-thrombi” in circulation is interesting. But since platelets are abundant in blood, how much heparin in H@CaPP could act on the “micro-thrombi”? Will this interaction between heparin and platelets lead to off-target effect and lead the nanotherapeutic to other places?

Response: Thanks for your questions. To form “micro-thrombi”, the platelets become activated by CD 44 and integrin α IIb β 3 on the surface of tumor cells^{14, 15}. Then, the activated platelets produced coagulation factors such as coagulation factor III to catalyze prothrombin into thrombin. Finally, thrombin contributes to platelets aberrant aggregation on the tumor cells and form the “micro-thrombi”^{16, 17}. The heparin inhibited the formation of “micro-thrombi” by hindering the conversion of prothrombin to thrombin rather directly bind to the activated platelets^{17, 18, 19}. In the case of H@CaPP, it could target to the area of platelets aberrant aggregation, such as the “micro-thrombi” or the site of inflammation, such as the pre-metastasis niche, through the specific interaction between LMWH and P-selectin (Figure 3), and then the nanoparticle could inhibit the formation of “micro-thrombi” and development of the pre-metastasis niche (Figure 7). Therefore, we believe H@CaPP would not largely influence the normal platelets in circulation and cause off-target effect. The detailed description about the formation of “micro-thrombi” has now been incorporated in the

revised manuscript, please see Line 332-340 (Page 16). Related results and analyses please see Line 340-357 (Page 16-17) and Line 376-399 (Page 18) in the revised manuscript.

Figure 3. H@CaPP targeted the pre-metastatic niche. **a** Uptake of D@CaPP and H@CaPP in HUVECs. DiI-labeled D@CaPP and H@CaPP (red) were incubated with a HUVECs pre-incubated with or without TNF- α (50 ng/mL, 4 h), respectively. In order to investigate the mechanism, an anti-P-selectin antibody was incubated with HUVECs for 2 h before adding nanotherapeutics. Afterwards, HUVECs on 6-wells plate were imaged after staining nuclei by honest 33258 (blue) for eight minutes. Scale, bar, 20 μ m. **b** Quantification of HUVECs uptake of nanotherapeutics under various conditions (ANOVA, n.s. indicated no significant difference (mean \pm SD, n = 3). **c** Expression of P-selectin in mice lungs examined by immunofluorescent staining. Normal mice and tumor-bearing mice were intravenously injected with DiI-labeled

nanotherapeutics (1 mg/kg DiI), respectively. Blue: Hoechst 33258 stained nuclei; green: P-selectin pre-metastatic sites; red: DiI-labeled NPs. Scale bar, 50 μm .

Figure 7. H@CaPP inhibited the formulation of pre-metastatic micro-environment *in vivo*. Representative images of immunohistochemical analysis of ICAM-1, VCAM-1, MMP-9 and S100A9 (brown) in lung of breast tumor-bearing mice after intravenously administrated with saline, LMWH, piceatannol, CaPP, D@CaPP or H@CaPP (*i.v.* at dose of piceatannol or PP at 0.020 mmol/kg, at dose of Dextran or LMWH at 10 mg/kg) for seven times in fourteen days. Scale bar, 50 μm .

6. Fig5a is an interesting *in vitro* study to prove the inhibitory effect of H@CaPP on the adhesion of platelets to tumor cells. Could the “micro-thrombi” be isolated or identified from the blood? Because the *in vitro* study is still not enough to prove the H@CaPP acts on the “circulation phase” but not later phases.

Response: Thank you for the comments. Indeed, it would be interesting to isolate or identify “micro-thrombi” in the blood circulation to prove the inhibitory effect of H@CaPP on the adhesion of platelets to tumor cells. However, current techniques can hardly identify “micro-thrombi” formed before or during the isolation process. Therefore, constructing the model *in vitro* is a main method for investigating the “micro-thrombi”. Hence, the adhesion assay of platelets to tumor cells was utilized for mimicking the formation of “micro-thrombi” *in vitro*. In order to provide more solid data, we performed the assay *in vitro*. Briefly, the platelets were obtained from whole blood of female BALB/c mice and labeled with calcein-AM. Then, the platelets were added to the 4T1 cells. After incubating for 30 min at 37°C with gentle shaking, the fluorescent intensity of platelets was quantified. Compared with PBS group, the fluorescence intensity of platelets in H@CaPP group were significantly

decreased 70.99%, suggesting H@CaPP could hinder the adhesion of platelets to tumor cells (Figure 5a-b). The detailed description about the method of adhesion assay in the revised manuscript, please see Line 768-776 (Page 35). The related results in the revised manuscript, please see Line 340-357 (Page 16-17).

Figure 5. H@CaPP inhibited the formation of “micro-thrombi” by inhibiting adhesion of platelets to tumor cells. **a** Adhesion of platelets to 4T1 cells, and the inhibitory effect of PBS, LMWH, piceatannol, D@CaPP and H@CaPP (dose of free piceatannol or PP at 10 μ M, dose of Dextran or LMWH at 1.3 mg/mL). The nuclei of 4T1 cells were stained with Hoechst 33258, and platelets were obtained and labeled with calcein-AM (green). Scale bar, 50 μ m. **b** The fluorescence intensity analysis of platelets adhering to 4T1 cells (n.s. indicates no significant difference with that of PBS, ANOVA, mean \pm SD, n = 5). **c** IHC analysis of CD-41 in tumors after intravenously administrated with saline, LMWH, piceatannol, D@CaPP and H@CaPP (*i.v.* at dose of piceatannol or PP at 0.020 mmol/kg, at dose of Dextran or LMWH at 10 mg/kg) for seven times in 14 days. Scale bar, 50 μ m.

7. Is Fig5c the primary tumor or metastatic tumor? The Figure shown in the H@CaPP group is strange since tumor as an inflammatory tissue is known for platelet aggregation and thrombi formation.

Response: Thank you for the question and comment. In order to investigate whether H@CaPP could inhibit adhesion of platelets to tumor cells *in vivo*, we performed the immunohistochemistry assay for CD41 in primary tumor (Figure 5c). As you mentioned, tumor is an inflammatory tissue containing platelets aberrant aggregation and thrombi formation. Figure 5c showed that platelets aggregation was less organized in H@CaPP group than other groups at primary tumor by IHC assay. The result supported that H@CaPP could inhibit the aberrant aggregation of platelets and formation of thrombi *in vivo*. Related description please see Line 347-357 (Page 16-17) in the revised manuscript.

8. Blood safety test (for example, hemolysis study) should be provided for this nanotherapeutic.

Response: Thank you for the suggestion. Accordingly, we performed additional hemolytic assay to study the blood safety of H@CaPP^{20,21}. The red blood cells were obtained from BALB/c mice. Samples incubated with 1% Triton X-100 served as the positive control and samples incubated with PBS as the negative control, respectively. Then, each sample was centrifuged at 3000 rpm for 15 min. The absorbance of supernatant was analyzed at 540 nm by a microplate reader. The abs value of H@CaPP did not show significant difference with that of the negative control (Figure S13), suggesting that H@CaPP possess negligible hemolytic toxicity. Related description please see Line 199-202 (Page 8) in the revised manuscript and the method has now been incorporated in the revised manuscript please see Line 739-744 (Page 34).

Figure S13. Hemolytic assay of H@CaPP. **a** The red blood cells incubated with PBS, H@CaPP and 1% Triton X-100 for 8 hours. **b** The abs value of red blood cells incubated with PBS, H@CaPP and 1% Triton X-100 for 8 hours. n.s. indicated no significant difference compared with PBS group, $P < 0.0001$ indicates significant difference compared with H@CaPP (ANOVA, means \pm SD, $n = 3$).

9. The title of this manuscript may be re-considered, since the emphasis of this study is using a nanotherapeutic to target the three starges of metastasis-but not reinforce surgery and chemotherapy.

Response: Thank you for the suggestion. The reviewer is right that here we focused on the design of a nanotherapeutic to target the three stages of metastasis. As the conventional clinical therapies such as tumor resection and chemotherapy are essential for treating tumors, but fail to inhibit or even induce metastasis, it's of great importance to develop an effective metastasis-specific therapeutic to serve as a complementary strategy to tumor resection and chemotherapy to combat tumor metastasis. Here, we justified that the designed nanotherapeutic could systemically inhibit tumor metastasis and reinforce surgery and chemotherapy. We believe the current title be of immediate interest to the broad audience of *Nature Communications*.

10. Page 5 line 135, there is no Scheme 1b in Scheme 1.

Response: Thank you very much for pointing out the mistake and we have corrected it accordingly. Please see Line 135 (Page 5) in the revised manuscript.

11. Fig S11a is interesting. Please clarify the n numbers in each group and time point. Is it reproducible and even in different mice?

Response: Thank you for the suggestions. We have clarified “n = 3” in revised Figure S14. In the experiment, we observed that the increase of inflammatory cells and the enhanced pre-metastasis-related markers (MMP-9 and S100A9) in the lungs tissue with the extension of tumor bearing time. We believe that the results could be reproducible in 4T1 tumor-bearing mice. Please see the caption of revised Figure S14 in the revised manuscript.

12. Page 10 line 243, I think the citation for Fig3d should Fig3c.

Response: Thank you very much for pointing out the mistake and we have corrected it accordingly. Please see Line 259 (Page 11) in the revised manuscript.

13. Page 17, line 388, “. And H@CaPP” is mistake.

Response: Thank you very much for pointing out the mistake and we have corrected it accordingly. Please see Line 388-389 (Page 18) in the revised manuscript.

Reviewer #2 (Remarks to the Author):

1. The authors claimed the ant-inflammatory effect is an important mechanism of their approach. However, no data on anti-inflammation of the H@CaPP nanoparticles is shown. Data on how the nanoparticle affects the hallmarks of inflammation (neutrophil and monocyte and their recruitment), should be provided. The mechanism of piceatannol on E-Cadherin, Vimentin, Vcam-1 and ICAM-1 are not provided or described clearly in this study either. The authors should provide more information on how piceatannol can regulate E-cadherin, vimentin, vcam-1 and ICAM-1 expression in 4T1 cells and ECs.

Response: Thank you for the comments and suggestions. In order to investigate how H@CaPP affects inflammation site, we have performed additional detailed analyses to further evaluate the recruitment of granulocytic myeloid derived suppressor cells (G-MDSCs) in the lung. The G-MDSCs could be recruited to inflammation sites, such as pre-metastatic niche, by multiple factors (VEGF, CXCL5, GM-CSF, IL-6, *etc.*) and can facilitate metastasis by causing the increased vascular permeability and immunosuppression^{22, 23, 24}. Ly6G is a marker of G-MDSC. Therefore, the expression Ly6G in lung was positively correlated with the recruitment of G-MDSCs to pre-metastatic niche²⁵. As shown in the result of immunohistochemistry assay (Figure S18), H@CaPP significantly exhibited inhibitory effects of Ly6G compared with other groups ($P < 0.0001$). The result supported that H@CaPP could hinder the recruitment of G-MDSCs and exerted anti-inflammation effect in the pre-metastatic niche. Related description please see Line 399-409 (Page 18-19) in the revised manuscript.

Piceatannol has been proved as a potential pharmaceutical ingredient in some inflammation-related diseases, such as arrhythmia, atherosclerosis and cardiovascular diseases^{26, 27}. It exhibits ant-inflammatory effect through multiple inflammatory pathways including the signal transducers and activators of transcription nuclear-3 (STAT-3) and nuclear factor- κ B (NF- κ B) pathways, etc. Vimentin and E-cadherin are two typical markers of epithelial-mesenchymal transition (EMT)^{28, 29}. The activation of STAT-3 could mediate EMT by reducing level of E-cadherin and enhancing level of vimentin³⁰. Piceatannol could suppress STAT3 phosphorylation and activation^{26, 31}. Therefore, piceatannol could reduce the expression of vimentin and evaluate the expression of E-cadherin to achieve inhibition of EMT process. VCAM-1 and ICAM-1 are adhesion molecules that over-expressed on endothelial cells response to inflammatory stimulation³². VCAM-1 and ICAM-1 are regulated by redox-sensitive transcription factors, in particular NF- κ B³³. Piceatannol could inhibit the activation of NF- κ B by suppressing the subunit p65 phosphorylation^{34, 35}. Therefore, piceatannol could reduce the expression of VCAM-1 and ICAM-1 on activated ECs. Consistent with previous findings, we found that piceatannol could inhibit EMT and hinder the aberrant expression of ICAM-1 and VCAM-1 (the revised Figure 4 and Figure 6).

Related results and analyses please see Line 288-311 (Page 14) and Line 381-389 (Page 18) in the revised manuscript.

Figure S18. H@CaPP inhibited recruitment of G-MDSCs to the pre-metastasis niche. **a** Representative image of immunological analyze of Ly6G (brown) in lung of breast tumor-bearing mice after intravenously administrated with saline, LMWH, piceatannol, CaPP, D@CaPP or H@CaPP (*i.v.* at dose of piceatannol or PP at 0.020 mmol/kg, at dose of Dextran or LMWH at 10 mg/kg) for seven times in fourteen days. Scale bar, 50 μ m. **b** Semi-quantitative analysis of Ly6G. Results were analyzed by ImageJ and presented as mean \pm SD. n = 10 section images. $P < 0.0001$ indicates significant difference compared with H@CaPP. Significant differences were assessed by using one-way ANOVA with multiple comparisons.

Figure 4. H@CaPP hindered invasion of primary tumor by inhibiting EMT process. **a** Images and **c** quantitative analysis of the number of invaded 4T1 cells after incubating tumor cells with PBS, LMWH, piceatannol, D@CaPP, or H@CaPP (dose of free piceatannol or PP at 10 μ M, dose of Dextran or LMWH at 1.3 mg/mL) for 24 h. The cells were stained with crystal violet and each purple spot represented one invaded cell at the field of vision (ANOVA, means \pm SD, n = 5). **b** Representative images of Western blot analysis of Vimentin and E-cadherin in 4T1 cells. **d** Quantitative analysis of the expression of Vimentin and **e** E-cadherin in 4T1 cells (ANOVA, means \pm SD, n = 3). **f** Representative images of immunological analysis of Vimentin and E-cadherin (brown) in tumors after intravenously administrated with saline,

LMWH, piceatannol, CaPP, D@CaPP and H@CaPP (*i.v.* at dose of piceatannol or PP at 0.020 mmol/kg, at dose of Dextran or LMWH at 10 mg/kg) for seven times in fourteen days. Scale bar, 50 μ m.

Figure 6. Evaluation of the adhesion of ECs in the pre-metastatic site *in vitro*. **a** Images of 4T1 cells (labeled calcein-AM) adhered to HUVECs which were treated

with PBS, TNF- α + PBS, TNF- α + LMWH, TNF- α + piceatannol, TNF- α + D@CaPP, and TNF- α + H@CaPP (TNF- α = 50 ng/mL, dose of free piceatannol or PP at 10 μ M, dose of Dextran or LMWH at 1.3 mg/mL) for 24 h, respectively. Left photo, fluorescent field; right photo, bright field. Scale bar, 50 μ m. **b** Number of 4T1 cells in per field from each group (ANOVA, mean \pm SD, n = 3). **c** Flow cytometry analysis of VCAM-1 and ICAM-1 on HUVECs treated as **a**, respectively. Cells without the incubation of antibodies were served as blank control. **d** Fluorescence intensity of VCAM-1 and ICAM-1 in each group (ANOVA, mean \pm SD, n = 3).

2. In Figure 5, they show the LMWH on the H@CaPP nanoparticles could inhibit the adhesion of platelets. Since platelets are abundant in the blood, how can the nanoparticles escape from binding to them in blood before reaching tumor? After the nanoparticles reach tumor and are taken up by activated ECs, how can they be effective in cancer cells and still regulate the expression of vimentin and E-cadherin?

Response: Thank you for the questions. To form “micro-thrombi”, the platelets become activated by CD44 and integrin α Ib β 3 on the surface of tumor cells^{14, 15}. Then, the activated platelets produced coagulation factors such as coagulation factor III to catalyze prothrombin into thrombin. Finally, thrombin contributes to the platelets aberrant aggregation on the tumor cells and form the “micro-thrombi”^{16, 17}. Heparin inhibited the formation of “micro-thrombi” by hindering the conversion of prothrombin to thrombin rather directly bind to the activated platelets^{17, 18, 19}. H@CaPP efficiently targeted the pre-metastatic niche that is an inflammation site via binding to P-selectin that expressed on the surface of activated ECs (Figure 3). And H@CaPP exhibited inhibitory effects of pre-metastatic niche development (Figure 7). Therefore, we believe H@CaPP would not largely influence the normal platelets in circulation and cause off-target effect. As to the primary tumor, H@CaPP would accumulate at the tumor site via enhanced permeability and retention effect (EPR effect) and then could target to tumor cells via binding to P-selectin^{36, 37, 38}. Then, it could achieve inhibitory effects on the EMT process (Figure 4). Related results and analyses please see Line 242-262 (Page 11), Line 295-311 (Page 14) and Line 389-412 (Page 18-19) in the revised manuscript. And the detailed description about the “micro-thrombi” has now been incorporated in the revised manuscript, please see Line 332-340 (Page 16).

Figure 3. H@CaPP targeted the pre-metastatic niche. **a** Uptake of D@CaPP and H@CaPP in HUVECs. DiI-labeled D@CaPP and H@CaPP (red) were incubated with a HUVECs pre-incubated with or without TNF- α (50 ng/mL, 4 h), respectively. In order to investigate the mechanism, an anti-P-selectin antibody was incubated with HUVECs for 2 h before adding nanotherapeutics. Afterwards, HUVECs on 6-wells plate were imaged after staining nuclei by honest 33258 (blue) for eight minutes. Scale, bar, 20 μ m. **b** Quantification of HUVECs uptake of nanotherapeutics under various conditions (ANOVA, n.s. indicated no significant difference (mean \pm SD, n = 3). **c** Expression of P-selectin in mice lungs examined by immunofluorescent staining. Normal mice and tumor-bearing mice were intravenously injected with DiI-labeled nanotherapeutics (1 mg/kg DiI), respectively. Blue: Hoechst 33258 stained nuclei; green: P-selectin pre-metastatic sites; red: DiI-labeled NPs. Scale bar, 50 μ m.

Figure 4. H@CaPP hindered invasion of primary tumor by inhibiting the EMT process. **a** Images and **c** quantitative analysis of the number of invaded 4T1 cells after incubating tumor cells with PBS, LMWH, piceatannol, D@CaPP, or H@CaPP (dose of free piceatannol or PP at 10 μ M, dose of Dextran or LMWH at 1.3 mg/mL) for 24 h. The cells were stained with crystal violet and each purple spot represented one invaded cell at the field of vision (ANOVA, means \pm SD, n = 5). **b** Representative images of Western blot analysis of Vimentin and E-cadherin in 4T1 cells. **d** Quantitative analysis of the expression of Vimentin and **e** E-cadherin in 4T1 cells (ANOVA, means \pm SD, n = 3). **f** Representative images of immunological analysis of Vimentin and E-cadherin (brown) in tumors after intravenously administrated with

saline, LMWH, piceatannol, CaPP, D@CaPP and H@CaPP (*i.v.* at dose of piceatannol or PP at 0.020 mmol/kg, at dose of Dextran or LMWH at 10 mg/kg) for seven times in fourteen days. Scale bar, 50 μ m.

Figure 7. H@CaPP inhibited the formulation of pre-metastatic micro-environment *in vivo*. Representative images of immunohistochemical analysis of ICAM-1, VCAM-1, MMP-9 and S100A9 (brown) in lung of breast tumor-bearing mice after intravenously administrated with saline, LMWH, piceatannol, CaPP, D@CaPP or H@CaPP (*i.v.* at dose of piceatannol or PP at 0.020 mmol/kg, at dose of Dextran or LMWH at 10 mg/kg) for seven times in fourteen days. Scale bar, 50 μ m.

3. The general knowledge is that heparin only makes blood thinner and decreases the clotting ability of blood as is also shown by the authors in Fig. 5, but it does not de-clot micro-thrombi already formed in blood. Since the authors claimed the anti-micro-thrombi ability of their H@CaPP nanoparticles, this should be demonstrated *in vitro*.

Response: Thanks for the valuable comments. Yes, LMWH could not de-clot micro-thrombi already formed in blood but it could hinder the conversion of prothrombin to thrombin via promoting antithrombin III (AT) activity^{18, 19}. Hence, the “anti-micro-thrombi ability” means LMWH could inhibit the aberrant aggregation of platelets on tumor cells^{17, 37, 39}. In our research, we have performed the adhesion assay *in vitro*. Briefly, the platelets were obtained from whole blood of female BALB/c mice and labeled with calcein-AM. Then, the platelets were added to the 4T1 cells. After incubating for 30 min at 37°C with gentle shaking, the fluorescent intensity of platelets was quantified. Both LMWH and H@CaPP could efficiently downregulate the formation of “micro-thrombi” via inhibiting adhesion of platelets to

tumor cells *in vitro* (Figure 5a-b). Related analyses of the results please see Line 340-357 (Page 16-17) in the revised manuscript.

Figure 5. H@CaPP inhibited the formation of “micro-thrombi” by inhibiting adhesion of platelets to tumor cells. **a** Adhesion of platelets to 4T1 cells, and the inhibitory effect of PBS, LMWH, piceatannol, D@CaPP and H@CaPP (dose of free piceatannol or PP at 10 μ M, dose of Dextran or LMWH at 1.3 mg/mL). The nuclei of 4T1 cells were stained with Hoechst 33258, and platelets were obtained and labeled with calcein-AM (green). Scale bar, 50 μ m. **b** The fluorescence intensity analysis of platelets adhering to 4T1 cells (n.s. indicates no significant difference with that of PBS, ANOVA, mean \pm SD, n = 5). **c** IHC analysis of CD-41 in tumors after intravenously administrated with saline, LMWH, piceatannol, D@CaPP and H@CaPP (*i.v.* at dose of piceatannol or PP at 0.020 mmol/kg, at dose of Dextran or LMWH at 10 mg/kg) for seven times in 14 days. Scale bar, 50 μ m.

4. The criteria for identifying pre-metastatic, invasion, micro-thrombi, and colonization phase are unclear at best. How can the authors specify the metastatic phase and do the treatment?

Response: Thanks for the comment and question. Metastasis includes the three main steps: the invasion, circulation and the colonization phases^{40, 41}. (i) In the invasion

phase, the tumor cells escape from the primary tumor tissue to surrounding tissues and extravasate into the neighboring blood vessels. The criterion of this phase is that the metastatic tumor grows rapidly and infiltrates the surrounding tissue⁴². (ii) In the circulation phase, the tumor cells that enter and survive in the bloodstream are called circulating tumor cells (CTCs)⁴³. The criterion of this phase is that platelets are bound to CTCs, forming the special structure called “micro-thrombi” that could prevent CTCs elimination from immune system^{44, 45}. (iii) In the colonization phase, CTCs infiltrate and colonize in the distant organs. The distant organs that provide suitable niche for CTCs colonization are defined as the pre-metastatic niche. The criterion of this phase is the formation of pre-metastatic niche^{40, 43}. In our study, we did the treatment according to the period of pre-metastatic niche formation because the formation of this niche is prerequisite for tumor metastasis. Therefore, in order to identify the period of pre-metastatic niche formation, lungs of tumor-bearing mice were resected and stained with hematoxylin-eosin staining (H&E) every second day to observe lung morphology. From day 4 to day 14 after tumor inoculation, changes in terms of lung morphology were observed in the tumor-bearing mice, with decrease in the number of alveoli and increase in the number of inflammatory cells. And the initial visible lung metastatic areas were observed at the 16th day after inoculation. (Figure S14a). To verify the formation of pre-metastatic niche in the lung, the expression S100A9 and MMP-9, two marker proteins of pre-metastatic niche^{46, 47}, were tested through immunohistochemistry (IHC). Compared with the normal mice group, lung section from tumor-bearing mice exhibited enhanced S100A9 and MMP-9 expression in the 14th day (Figure S14b). Hence, according to results of the preliminary experiment, the period of pre-metastatic niche formation would be appeared at early primary tumor development and be positively correlated with the tumor growth. Based on the experimental results, we designed the treatment plan for tumor-bearing mice. The mice were intravenously administrated with the nanotherapeutic in fourteen days after tumor cells inoculation. Related description of “the invasion, circulation and the colonization phases”, please see Line 78-101 (Page 3-4) in the revised manuscript. Related description of identifying the period of pre-metastatic niche formation in “H@CaPP targeted primary tumor and pre-metastatic site” section, please see Line 215-230 (Page 10) in the revised manuscript.

Figure S14. Construction of the pre-metastatic model in the orthotopic 4T1 cells-bearing mice. **a** Respective H&E images of the lungs tissue from normal mice or tumor-bearing mice at 2-16 days after inoculation. Black arrow indicates aberrant aggregation of inflammatory cells. Black circle indicates metastasis area. Scale bar, 100 μm . **b** Expression of MMP-9 and S100A-9 (brown) in the lungs from normal mice or tumor-bearing mice at 4 day and 14 day after inoculation. Scale bar, 50 μm . There were three mice for analysis in each time point (n=3).

5. The authors assumed that by targeting multiple steps of the EMT process, they can effectively reduce/inhibit cancer metastasis. In a recent study by Fischer et al (doi: 10.1038/nature15748), it is shown that EMT inhibition in lung cancer did not impact lung metastasis, and instead, it contributes to the chemoresistance. This contradicts the authors' claim of targeting the EMT for reducing lung metastasis. Although the authors show the data of the H@CaPP accumulation into the lungs and tumor, they do not show any data to prove that once the EMT process is hindered, the metastatic tumor cells become non-metastatic.

Response: Thanks for your comments. It has been widely recognized that EMT process is highly associated with the conversion of stationary cells to motile cells and is essential for invasion of primary tumor cells^{48, 49}. Therefore, we believe that the EMT process of tumor cells is the key step for tumor metastasis and inhibition of the

process is of paramount importance for anti-metastasis. In our study, we investigated whether the nanoparticles could hinder tumor cell invasion by inhibiting EMT. We performed the invasion assay and western blot for verifying that H@CaPP could inhibit tumor cells invasion and hinder EMT process *in vitro*. In addition, the IHC assay was utilized to evaluate inhibition of EMT process *in vivo*. Once the EMT process is hindered by H@CaPP, the invasiveness of metastatic tumor cells become lower than normal tumor cells (Figure 4a, c). Next, we investigated the therapeutic effect of H@CaPP combined with chemotherapy (free PTX) in spontaneous breast-to-lung metastasis models. The data demonstrated the nanotherapeutic could increase the therapeutic efficacy of chemotherapy for anti-metastasis (Figure 8). Hence, we believe that the metastatic tumor cells would become non-metastatic via inhibition of EMT and inhibition of the process has clinical potential for anti-metastasis. Fischer's article provokes our thinking of the epithelial-mesenchymal transition (EMT) process in tumor metastasis. Fischer and his colleagues established an EMT lineage-tracing system to monitor the process in mice, using a mesenchymal-specific Cre-mediated fluorescent marker switch system in spontaneous breast-to-lung metastasis models and inhibit EMT by overexpressing the microRNA miR-200. They claimed that lung metastases mainly consist of non-EMT tumor cells that maintain their epithelial phenotype and inhibiting EMT by overexpressing the miR-200 does not affect lung metastasis development. However, hindering EMT process could abrogate the chemoresistance contributing to inhibitory of recurrent lung metastasis formation after chemotherapy⁵⁰. Hence, the study suggested the potential of an EMT-targeting strategy, in conjunction with conventional chemotherapies, for breast cancer treatment. Obviously, both Fischer and we agree that inhibition of EMT would increase the therapeutic efficacy of chemotherapy for combating metastatic tumor. In Fischer's study, they focused on the cells in lung metastasis and claim that the cells would maintain their epithelial phenotype. In our research, we focused on the EMT process in primary tumor and verified that the invasiveness of tumor cells would be decreased by inhibition of EMT. During the whole process of tumor metastasis, the mesenchymal phenotype cells are important for intravasation and extravasation but the cells may transit to epithelial type for promoting their proliferation and growth when they colonized in the distant organs. Hence, we believe that inhibition of EMT in primary tumor could hinder the process of metastasis and reinforce the therapeutic effect of chemotherapy for anti-metastasis. Related the results and the analyses, please see Line 288-311 (Page 14) and Line 435-453 (Page 21-22) in the revised manuscript.

Figure 4. H@CaPP hindered invasion of primary tumor by inhibiting EMT process. **a** Images and **c** quantitative analysis of the number of invaded 4T1 cells after incubating tumor cells with PBS, LMWH, piceatannol, D@CaPP, or H@CaPP (dose of free piceatannol or PP at 10 μ M, dose of Dextran or LMWH at 1.3 mg/mL) for 24 h. The cells were stained with crystal violet and each purple spot represented one invaded cell at the field of vision (ANOVA, means \pm SD, n = 5). **b** Representative images of Western blot analysis of Vimentin and E-cadherin in 4T1 cells. **d** Quantitative analysis of the expression of Vimentin and **e** E-cadherin in 4T1 cells (ANOVA, means \pm SD, n = 3). **f** Representative images of immunological analysis of Vimentin and E-cadherin (brown) in tumors after intravenously administrated with saline,

LMWH, piceatannol, CaPP, D@CaPP and H@CaPP (*i.v.* at dose of piceatannol or PP at 0.020 mmol/kg, at dose of Dextran or LMWH at 10 mg/kg) for seven times in fourteen days. Scale bar, 50 μ m.

Figure 8. H@CaPP combined with chemotherapy achieved anti-metastasis therapeutic effect in orthotopic primary tumor-bearing mice. **a** Typical lung tissues with visualized metastatic nodules (white arrows) from each group, and representative fields of lung tissues with metastasis areas (blue dotted lines) from each group with H&E. Black scale bar, 5 mm; White scale bar, 0.5 mm. **b** The number of metastatic nodules from the mice after intravenously administration with saline, PTX, PTX + LMWH, PTX + piceatannol, PTX + D@CaPP or PTX + H@CaPP (*i.v.* of PTX at 5.9×10^{-3} mmol/kg, at dose of piceatannol or PP at 0.020 mmol/kg, at dose of Dextran or LMWH at 10 mg/kg) for seven times in fourteen days (ANOVA, mean \pm SD, n = 5). **c** Survival curve of orthotopic breast tumor-bearing mice combined PTX after intravenously administration with saline, PTX, PTX + LMWH, PTX + piceatannol, PTX + D@CaPP or PTX + H@CaPP for seven times during fourteen days (n=6).

6. To maintain the multifunctional efficacy in different tumor phase and different cells, pharmacokinetics of LMWH and piceatannol from the H@CaPP nanoparticles should be provided. Also, the *in vitro* release profile at tumor pH should be included.

Response: Thanks for the valuable suggestions. Accordingly, we have performed additional detailed analyses to further evaluate pharmacokinetics of LMWH and piceatannol from the H@CaPP. Pharmacokinetics of LMWH and the formulation were evaluated by anti-FXa assay in SD rats^{51, 52}. The obtained data were analyzed with the main pharmacokinetic parameters (including AUC_{0-t}, k, t_{1/2}, Cl) and the pharmacokinetic profiles presented in Table S3 and Figure S23, respectively. After loaded on the surface of the PEGylated nanoparticle (H@CaPP), the half-life blood circulation time of LWMH was obviously prolonged. Compared with LMWH (t_{1/2} 2.71h), H@CaPP exhibited longer half-life blood circulation time (t_{1/2} 4.09h). In addition, LWMH coated on CaPP showed slower clearance (Cl) rate and higher AUC_{0-t} (about 2.1 folds) when compared with free LMWH. The results demonstrated that H@CaPP exhibited a superior blood circulation profile compared with LMWH. As to piceatannol, high performance liquid chromatography/electrospray ionization mass spectrometry method has been developed for the simultaneous determination of piceatannol phosphate (PP) and piceatannol (PIC) in rat plasma. The Triple Quad™ 6500 triple-quadrupole tandem mass spectrometry (SCIEX, USA) was operated under the multiple reaction-monitoring mode (MRM) using the negative electrospray ionization technique. As the results shown in Table S4 and Figure S24, the mean AUC_{0-t} (nmol/ml/h) of PIC, PP, CaPP and H@CaPP were 2.87, 2.03, 7.51 and 16.59, respectively. And H@CaPP (7.82h) exhibited obviously longer half-life blood circulation time compared with PIC (0.71h), PP (0.48h) and CaPP (3.71h). In contrast, PIC and PP were all cleared the fastest as expected. CaPP was cleared more quickly than H@CaPP probably because of its recognition by the mononuclear phagocyte system (MPS) due to the surface positive charge^{53, 54}. These data collectively demonstrated that LMWH modification improved the blood circulation profile of CaPP, and the long-term blood circulation of nanotherapeutic would ensure better targeting effects to whole process of metastasis. Related description please see Line 529-552 (Page 26-27) and the methods of pharmacokinetics has now been incorporated please see Line 882-912 (Page 39-40) in the revised manuscript.

We have also further studied the release profiles of PP from H@CaPP at the tumor intracellular conditions. As reported, the calcium-phosphate liposome would be internalized by endosomal/lysosomal pathway^{11, 12, 13}. Therefore, we evaluated the PP release of H@CaPP in phosphate buffer solution at pH 6.5 (mimicking endosomes) and 5.0 (mimicking lysosomes). H@CaPP was packaged in the dialysis bags (MWCO 13000 Da) under gentle shaking. At various time points (0, 2, 4, 8, 12, 24 h), the amount of PP released was determined by the HPLC analysis. In pH of lysosomes (5.0), the H@CaPP released 83.0% of PP in 24 h. Whereas, in pH of lysosomes (6.5), only 35.5% of PP was released from the H@CaPP in 24 h. The result indicated that the release rate of PP was significantly dependent on the pH value and the drug was released from lysosomes into cytoplasm. Related description please see Line 182-190 (Page 7) in the revised manuscript and the method has now been incorporated in the revised manuscript please see Line 732-738 (Page 34).

Figure 23 Anti-FXa activity vs time profiles of LMWH solution and H@CaPP after intravenous administration in equivalent dose of 100 IU/kg in SD rats. Values represent mean \pm SD (n = 3).

Table S3. Pharmacokinetic parameters of LMWH formulations after intravenous administration of LMWH solution and H@CaPP in SD rats. * $p < 0.05$ or ** $p < 0.005$ indicates significant difference (t -test, mean \pm SD, n=3).

Formulations	AUC _{0-t} (IU/ml/h)	k (h ⁻¹)	t _{1/2} (h)	Cl (ml/h/kg)
LMWH	1.82 \pm 0.22	0.26 \pm 0.04	2.71 \pm 0.46	53.29 \pm 5.59
H@CaPP	3.88 \pm 0.94*	0.17 \pm 0.01*	4.09 \pm 0.36*	29.54 \pm 3.30**

Figure S24. Plasma piceatannol or piceatannol phosphate concentration-time curves after intravenously administration of free piceatannol, free piceatannol phosphate,

CaPP, H@CaPP respectively, at an equivalent piceatannol dose of 0.02 mmol/kg in SD rats, n = 3. Data presented means \pm SD.

Table S4. Pharmacokinetic parameters of the PIC/PP after intravenous administration of PIC solution, PP solution, CaPP and H@CaPP in SD rats. * $P < 0.05$, ** $P < 0.005$, *** $P < 0.0005$ and **** $P < 0.0001$ indicates significant difference compared with H@CaPP (ANOVA, mean \pm SD, n=3).

Formulations	AUC _{0-t} (nmol/ml/h)	k (h ⁻¹)	t _{1/2} (h)	Cl (ml/h/kg)
PIC	2.64 \pm 0.54**	1.11 \pm 0.38	0.71 \pm 0.26*	7.64 \pm 1.44***
PP	2.03 \pm 0.07***	2.06 \pm 0.98*	0.48 \pm 0.31*	9.02 \pm 0.67****
CaPP	7.51 \pm 3.45**	0.22 \pm 0.09	3.71 \pm 1.33	2.53 \pm 0.91
H@CaPP	16.59 \pm 2.79	0.11 \pm 0.04	7.82 \pm 3.64	1.06 \pm 0.13

Figure S10. The PP release profile of H@CaPP in different pH medium (5.0 or 6.5) within 24 h (mean \pm SD, n = 3).

7. In the discussion part, the authors mentioned that PP could be dephosphorylated by phosphatases that are extensively expressed by cancer cells. How was the expression of phosphatases in activated ECs and 4T1 cells?

Response: Thanks for the question. Phosphatase exists extensively on cell membranes, cytoplasm, and lysosomes, especially in tumor cells^{55, 56}. It has been reported that the enzyme is elevated in the cells of tumor metastasis. For example, the seminoma marker Regan isoenzyme of alkaline phosphatase in human myeloma cells is considered as a metastasis-associated phosphatase^{57, 58}. It is likely that the upregulated phosphatase in the tumor metastasis may result in an enhanced conversion of PP to piceatannol. As to the two mentioned cells, the 4T1 cell is a tumor cell and the activated EC is a metastasis-associated cell. Hence, the two cells would highly express phosphatase. In our research, as shown in Figure S11, we successfully detected the intracellular conversion of PP to piceatannol in 4T1 cells and activated ECs, which indicated that the phosphatases are expressed in the cells.

The relevant results and analyses please see Line 190-193 (Page 7-8) in the revised manuscript.

Figure S11. Conversion of PP to piceatannol in 4T1/HUVEC cells. **a** Determination of piceatannol in the cell medium after incubating 4T1/HUVEC cells with H@CaPP at 37°C for 4 h. **b** HPLC spectrum of piceatannol standard. **c** HPLC spectrum of the 4T1 cells culture medium. **d** HPLC spectrum of the 4T1 cells culture medium after incubating 4T1 cells with H@CaPP at 37°C for 4 h. **e** HPLC spectrum of PP standard. **f** HPLC spectrum of the HUVEC cells culture medium. **g** HPLC spectrum of the HUVEC cells culture medium after incubating HUVEC cells with H@CaPP at 37°C for 4 h.

8. The schematic illustration is unclear. More explanation is needed in the caption to explain what is going on in the schematic including the three targeting phases.

Response: Thanks for the valuable suggestion. We agree that more explanation should be supplied in Scheme 1. We have added the three target sites of H@CaPP in the caption of revised Scheme 1.

Scheme 1. Schematic illustration of the composition and anti-metastasis function of H@CaPP. H@CaPP was intravenously administered to suppress the whole metastasis process. H@CaPP hindered invasion phase via inhibition EMT process in primary tumor, hindered colonization phase via inhibiting formation of “micro- thrombi” in the circulation, and hindered the colonization phase via targeting P-selectin in the pre-metastatic niche and inhibiting typical aberrant alterations in the niche.

9. The authors should explain more on the low radiant efficiency of the H@CaPP in the tumor/area in Fig. 2d. The image in Fig. 2c clearly shows that the ex vivo fluorescence of the tumor is higher than the rest according to the intensity bar. There is also a confusion in Fig. 4b, where the western blot data does not match the caption.

Response: Thanks for the careful checking. The confusions were caused by editing mistakes in both Figure 2 and Figure 4. In Figure 2, the labels of intensity bar were wrong. The “low” should be marked on the red end and the “low” should be marked on the blue end of the bar. We have corrected the mistakes in revised Figure 2. As shown in the revised Figure 2, there is high radiant efficiency of the H@CaPP in the tumor site. The caption of Figure 4 has been corrected. The caption of Figure 4c should be “quantitative analysis of the number of invaded 4T1 cells after incubating tumor cells with PBS, LMWH, piceatannol, D@CaPP, or H@CaPP.” And the caption of Figure 4b should be “representative image of western blot analysis of Vimentin and E-cadherin in 4T1 cells”. We have corrected the mistakes in revised Figure S2 and in the caption of revised Figure 4 in the revised manuscript.

Figure 2. H@CaPP targeted primary tumor and pre-metastatic site *in vivo*. **a** *In vivo* fluorescence imaging of whole body of orthotopic tumor-bearing mice at 2, 4, 8, 12 and 24 h after intravenous injection of free DiR, DiR-labeled D@CaPP and H@CaPP at the dose of 1 mg/kg DiR. The black cycles indicate the sites of tumors and lungs. **b** Fluorescence imaging of lung of tumor-bearing mice at 2, 4, 8, 12 and 24 h after intravenous injection with free DiR-labeled D@CaPP and H@CaPP at the dose of 1 mg/kg DiR. Black cardboards were utilized to block fluorescence signals from tumors and other organs. **c** Images of the organs and tumors of mice in each group at 24 h. **d** Semi-quantitative analysis of fluorescence intensity of the *ex vivo* organs and tumors (mean ± SD, n = 3).

Figure 4. H@CaPP hindered invasion of primary tumor by inhibiting EMT process. **a** Images and **c** quantitative analysis of the number of invaded 4T1 cells after incubating tumor cells with PBS, LMWH, piceatannol, D@CaPP, or H@CaPP (dose of free piceatannol or PP at 10 μ M, dose of Dextran or LMWH at 1.3 mg/mL) for 24 h. The cells were stained with crystal violet and each purple spot represented one invaded cell at the field of vision (ANOVA, means \pm SD, n = 5). **b** Representative images of Western blot analysis of Vimentin and E-cadherin in 4T1 cells. **d** Quantitative analysis of the expression of Vimentin and **e** E-cadherin in 4T1 cells (ANOVA, means \pm SD, n = 3). **f** Representative images of immunological analysis of Vimentin and E-cadherin (brown) in tumors after intravenously administrated with saline,

LMWH, piceatannol, CaPP, D@CaPP and H@CaPP (*i.v.* at dose of piceatannol or PP at 0.020 mmol/kg, at dose of Dextran or LMWH at 10 mg/kg) for seven times in fourteen days. Scale bar, 50 μm .

10. In Fig. 7, there is only a minor exhibition of VCAM-1 inhibition by H@CaPP and an insignificant inhibition of ICAM-1. Please clarify.

Response: Thanks for your comments. The result in Figure 7 is not obvious could be attributed to the relative low levels of ICAM-1 and VCAM-1 in lung^{59,60}. To be more accurate, we conducted the semi-quantitative analysis of the ICAM-1 and VCAM-1 levels and supplemented it in Figure S17. The result showed that H@CaPP could inhibit the expression of ICAM-1 and VCAM-1 compared with other groups *in vivo*, although in a less extent than that of S100A9 ($P < 0.0001$) and MMP-9 ($P < 0.0001$) (Figure 7, Figure S17). In addition, H@CaPP can hinder the expression of ICAM-1 and VCAM-1 on activated ECs *in vitro* (Figure 6). Taken together, these data supported that H@CaPP could effectively inhibit the development of pre-metastatic niche. The semi-quantitative analysis of VCAM-1, ICAM-1, S100A9 and MMP-9 have been added to supplementary information (Figure S17) and related analyses of the results please see Line 383-412 (Page 18-19) in the revised manuscript.

Figure 6. Evaluation of the adhesion of ECs in the pre-metastatic site *in vitro*. **a** Images of 4T1 cells (labeled calcein-AM) adhered to HUVECs which were treated with PBS, TNF- α + PBS, TNF- α + LMWH, TNF- α + piceatannol, TNF- α + D@CaPP, and TNF- α + H@CaPP (TNF- α = 50 ng/mL, dose of free piceatannol or PP at 10 μ M, dose of Dextran or LMWH at 1.3 mg/mL) for 24 h, respectively. Left photo, fluorescent field; right photo, bright field. Scale bar, 50 μ m. **b** Number of 4T1 cells in

per field from each group (ANOVA, mean \pm SD, n = 3). **c** Flow cytometry analysis of VCAM-1 and ICAM-1 on HUVECs treated as **a**, respectively. Cells without the incubation of antibodies were served as blank control. **d** The fluorescence intensity of VCAM-1 and ICAM-1 in each group (ANOVA, mean \pm SD, n = 3).

Figure 7. H@CaPP inhibited the formulation of pre-metastatic micro-environment *in vivo*. Representative images of immunohistochemical analysis of ICAM-1, VCAM-1, MMP-9 and S100A9 (brown) in lung of breast tumor-bearing mice after intravenously administrated with saline, LMWH, piceatannol, CaPP, D@CaPP or H@CaPP (*i.v.* at dose of piceatannol or PP at 0.020 mmol/kg, at dose of Dextran or LMWH at 10 mg/kg) for seven times in fourteen days. Scale bar, 50 μ m.

Figure S17. Semi-quantitative analyses of VACM-1, ICAM-1, S100A9 and MMP-9 in Figure 7. Results were analyzed by ImageJ and presented as mean \pm SD. $n = 10$ section images. Significant differences were assessed by using one-way ANOVA with multiple comparisons.

11. The authors should review carefully and fix many grammatical errors such as those in (391-392), (410), 439, (449-450), Fig.2d (kidney?), Fig S14 (92); they should correct the caption in line 399-400 (Left photo is fluorescence image, not bright field as they stated) and proofread the manuscript carefully. The authors also mentioned the use of AVONA in (Fig. 4, 313 &317) for their statistical analysis. I believe it should be ANOVA. Number of mice per group for all group (e.g. survival experiments) should be clarified. Also, the manuscript would benefit a lot from a careful proofreading by a native English speaker.

Response: Thank you very much for pointing out the mistake and we have corrected it accordingly. Number of mice per group has been added in survival experiments (Figure 8c and 9c). We have asked a native speaker to modify and improve the language for the manuscript.

Reviewer #3 (Remarks to the Author):

1- In the Figure 2, there is a discrepancy between Figures c and d regarding fluorescence of D@CaPP and H@CaPP in the kidney.

Response: Thanks for the careful checking. The discrepancy was caused by editing mistakes in Figure 2. The labels of intensity bar were wrong. The “low” should be marked on the red end and the “low” should be marked on the blue end of the bar. We have corrected the mistakes in revised Figure 2.

Figure 2. H@CaPP targeted primary tumor and pre-metastatic site *in vivo*. **a** *In vivo* fluorescence imaging of whole body of orthotopic tumor-bearing mice at 2, 4, 8, 12 and 24 h after intravenous injection of free DiR, DiR-labeled D@CaPP and H@CaPP at the dose of 1 mg/kg DiR. The black cycles indicate the sites of tumors and lungs. **b** Fluorescence imaging of lung of tumor-bearing mice at 2, 4, 8, 12 and 24 h after intravenous injection with free DiR-labeled D@CaPP and H@CaPP at the dose of 1 mg/kg DiR. Black cardboards were utilized to block fluorescence signals from tumors and other organs. **c** Images of the organs and tumors of mice in each group at 24 h. **d** Semi-quantitative analysis of fluorescence intensity of the *ex vivo* organs and tumors (mean \pm SD, n = 3).

2- Looking at IHC slides of the Figure 4F, it is not clear how real difference could be noted between CaPP and H@CaPP. The authors may therefore explain which quantification has been performed? A similar issue on quantification is raised for the Figure 7.

Response: Thanks for the question and suggestion. To be more accurate, we have added the results of semi-quantitative analyses based on the Figure 4f and Figure 7. As shown in revised Figure S16, H@CaPP could decrease the expression of vimentin in primary tumor and increase the expression of E-Cadherin in primary tumor, compared to other groups ($P < 0.0001$). The results demonstrated H@CaPP could effectively inhibit EMT *in vivo*. As shown in Figure S17, H@CaPP exhibited significantly inhibitory effects of VCAM-1, ICAM-1, S100A9, and MMP-9 compared with other groups. Thus, H@CaPP could inhibit development of pre-metastatic niche. The semi-quantitative analyses of Vimentin, E-cadherin, VACM-1, ICAM-1, S100A9 and MMP-9 have been added to supplementary information and related analyses of the results please see Line 303-306 (Page 14) and Line 389-399 (Page 18-19) in the revised manuscript.

Figure S16. Semi-quantitative analyses of Vimentin and E-Cadherin in Figure 4f. Results were analyzed by ImageJ and presented as mean \pm SD. $n = 10$ section images. $P < 0.0001$ indicates significant difference compared with H@CaPP. Significant differences were assessed by using one-way ANOVA with multiple comparisons.

Figure S17. Semi-quantitative analyses of VACM-1, ICAM-1, S100A9 and MMP-9 in Figure 7a. Results were analyzed by ImageJ and presented as mean \pm SD. n = 10 section images. Significant differences were assessed by using one-way ANOVA with multiple comparisons.

3- Concerning in vivo survival data, it is not clear how the authors have exactly processed. In the M&M section, it is mentioned that « the 60-days survival period of each group... ; line 753) has been checked. It might be explain, in such a situation, how the authors were able to follow mice survival and concomitantly obtain lung materials available for metastatic assessment? Indeed, they should have been confronted to the issue that some mice dead before any lung obtention. The paper did not mention the number of mice per group, but we can define it between 5 or 6. But, how many lungs have been studied per group? We have not the information. Yet, this point is important because it is well known that there is heterogeneity between mice for which a sufficient number of animal is required for any molecular study (at least 3 mice).

Response: Thanks for the questions and the valuable comments. The number of obtained lungs is five per group (n=5) and the number of mice in survival experiment

is six per group (n=6). Sixteen days after the last administration, five mice were euthanized and lung of the mice were excised for evaluating the anti-metastasis effect. According to our observation and record, there is no orthotropic breast tumor-bearing mice dead from lung metastasis at the sixteenth day after the last administration. The six mice were monitored continuously and the survival time was recorded. We have clarified the number of animals in the caption of revised Figure 8 and Figure 9.

Figure 8. H@CaPP combined with chemotherapy achieved anti-metastasis therapeutic effect in orthotropic primary tumor-bearing mice. **a** Typical lung tissues with visualized metastatic nodules (white arrows) from each group, and representative fields of lung tissues with metastasis areas (blue dotted lines) from each group with H&E. Black scale bar, 5 mm; White scale bar, 0.5 mm. **b** The number of metastatic nodules from the mice after intravenously administration with saline, PTX, PTX + LMWH, PTX + piceatannol, PTX + D@CaPP or PTX + H@CaPP (*i.v.* of PTX at 5.9×10^{-3} mmol/kg, at dose of piceatannol or PP at 0.020 mmol/kg, at dose of Dextran or LMWH at 10 mg/kg) for seven times in fourteen days (ANOVA, mean \pm SD, n = 5). **c** Survival curve of orthotropic breast tumor-bearing mice combined PTX after intravenously administration with saline, PTX, PTX + LMWH, PTX + piceatannol, PTX + D@CaPP or PTX + H@CaPP for seven times during fourteen days (n=6).

Figure 9. H@CaPP combined with surgery resection achieved anti-metastasis therapeutic effect in orthotropic primary tumor-bearing mice. **a** Typical lung tissues with visualized metastatic nodules (white arrows) from each group, and representative fields of lung tissues with metastasis areas (blue dotted lines) from each group with H&E. Black scale bar, 5 mm; White scale bar, 0.5 mm. **b** The number of metastatic nodules of the mice combined with surgical resection after intravenously administration with saline (without surgical resection), saline, LMWH, piceatannol, D@CaPP or H@CaPP (*i.v.* at dose of piceatannol or PP at 0.020 mmol/kg, at dose of Dextran or LMWH at 10 mg/kg) for seven times during fourteen days (ANOVA, mean \pm SD, n = 5). **c** Survival curve of orthotropic breast tumor-bearing mice combined surgery after intravenous administration with saline (without surgical resection), saline, LMWH, piceatannol, D@CaPP or H@CaPP for seven times during fourteen days (n=6).

4- In the *in vivo* studies (Figures 8 and 9), histopathological analyses only concern lung and the occurrence of metastases. It might be useful to add tumor study (in terms of EMT) and lungs (in terms of metastatic niches as well as various markers previously studied in the paper, i.e. E-selectin, P-selectin, CD41, VCAM-1, ICAM-1, S100A9, and MMP9). Those results may definitively confirmed the anti-metastatic activity of H@CaPP.

Response: Thanks for the suggestion. We agree with the reviewer that the characterizations of above makers *in vivo* are important. Therefore, we analyzed the expression of Vimentin, E-Cadherin (Figure 4f-g) and CD41(Figure 5c) in tumor and the expression of E-selectin, P-selectin (Figure S15), VCAM-1, ICAM-1, S100A9, and MMP-9 (Figure 7) in lung after intravenously administrated the nanotherapeutic. In the studies of therapeutic effects of H@CaPP combined with chemotherapy and surgery resection (Figure 8 and 9), we aimed to evaluate the therapeutic effect and clinical potential of the nanotherapeutic. Thus, we did not repeat the characterizations of the above markers *in vivo* to avoid additional sacrifice of the experimental animals.

Figure 4. H@CaPP hindered invasion of primary tumor by inhibiting EMT process. **a** Images and **c** quantitative analysis of the number of invaded 4T1 cells after incubating tumor cells with PBS, LMWH, piceatannol, D@CaPP, or H@CaPP (dose of free piceatannol or PP at 10 μ M, dose of Dextran or LMWH at 1.3 mg/mL) for 24 h. The cells were stained with crystal violet and each purple spot represented one invaded cell at the field of vision (ANOVA, means \pm SD, n = 5). **b** Representative images of Western blot analysis of Vimentin and E-cadherin in 4T1 cells. **d** Quantitative analysis of the expression of Vimentin and **e** E-cadherin in 4T1 cells (ANOVA, means \pm SD, n = 3). **f** Representative images of immunological analysis of Vimentin and E-cadherin (brown) in tumors after intravenously administrated with saline, LMWH, piceatannol, CaPP, D@CaPP and H@CaPP (*i.v.* at dose of piceatannol or PP at 0.020 mmol/kg, at dose of Dextran or LMWH at 10 mg/kg) for seven times in fourteen days. Scale bar, 50 μ m.

Figure 5. H@CaPP inhibited the formation of “micro-thrombi” by inhibiting adhesion of platelets to tumor cells. **a** Adhesion of platelets to 4T1 cells, and the inhibitory effect of PBS, LMWH, piceatannol, D@CaPP and H@CaPP (dose of free piceatannol or PP at 10 μ M, dose of Dextran or LMWH at 1.3 mg/mL). The nuclei of 4T1 cells were stained with Hoechst 33258, and platelets were obtained and labeled with calcein-AM (green). Scale bar, 50 μ m. **b** The fluorescence intensity analysis of

platelets adhering to 4T1 cells (n.s. indicates no significant difference with that of PBS, ANOVA, mean \pm SD, n = 3). **c** IHC analysis of CD-41 in tumors after intravenously administrated with saline, LMWH, piceatannol, D@CaPP and H@CaPP (*i.v.* at dose of piceatannol or PP at 0.020 mmol/kg, at dose of Dextran or LMWH at 10 mg/kg) for seven times in 14 days. Scale bar, 50 μ m.

Figure 7. H@CaPP inhibited the formation of pre-metastatic micro-environment *in vivo*. Representative images of immunohistochemical analysis of ICAM-1, VCAM-1, MMP-9 and S100A9 (brown) in lung of breast tumor-bearing mice after intravenously administrated with saline, LMWH, piceatannol, CaPP, D@CaPP or H@CaPP (*i.v.* at dose of piceatannol or PP at 0.020 mmol/kg, at dose of Dextran or LMWH at 10 mg/kg) for seven times in fourteen days. Scale bar, 50 μ m.

Figure S15. Elevated expression of E-selectin and P-selectin in the activated ECs. **a** Respective immunofluorescent image of the expression of E-selectin in the lungs from normal mice or orthotopic breast tumor-bearing mice. Green, E-selectin; blue, Hoechst 33258, nuclear stain. Scale bar, 50 μ m. **b** Flow cytometry analysis and the fluorescence intensity of E-selectin on HUVECs which were treated with PBS or TNF- α . (*t*-test, mean \pm SD, *n*=3). **c** Respective immunofluorescent images of the expression of P-selectin in the lungs from normal mice or orthotopic breast tumor-bearing mice. Green, P-selectin; blue, Hoechst 33258, nuclear stain. Scale bar, 50 μ m. **d** Flow cytometry analysis and the fluorescence intensity of P-selectin on HUVECs which were treated with PBS or TNF- α . (*t*-test, mean \pm SD, *n*=3).

5- Finally, in the toxicity assessment, it lacks determination of hepatic, hematological and renal data. Moreover, the Figure 10 might be moved to the Supplementary section.

Response: Thanks for the suggestion. Accordingly, we have performed additional detailed analyses to further evaluate the potential toxicity of the formulation. We collected peripheral blood for hematological and serum biochemical analyses. As shown in Figure S22 a-c, there were no significant downward trends of WBC (white

blood cell,) RBC (red blood cell) counts and PLT (platelets) counts in the H@CaPP-treated group compared with the PBS group. This result suggested that the nanotherapeutic did not cause side effects on the immune system. The biochemical analysis of the serum was also conducted to evaluate organ toxicity (Figure S22 d-g). H@CaPP did not increase the levels of ALT (alanine aminotransferase), AST (aspartate trans-aminase), BUN (blood urea nitrogen) and CRE (creatinine), which indicated that the nanotherapeutic would not cause liver injury and kidney injury. Hence, the safety evaluation demonstrated that the nanotherapeutic were biocompatible and had clinical transformation potential. And the all figures of “Safety Evaluation *in vivo*” section have been moved to the Supplementary in revised manuscript. Related description please see Line 516-527 (Page 26) in the revised manuscript and the method has now been incorporated in the revised manuscript please see Line 879-881 (Page 39).

Figure S22. Blood routine examination and serum biochemistry data of mice intravenously administrated with saline, LMWH, piceatannol, CaPP, D@CaPP or H@CaPP (*i.v.* at dose of piceatannol or PP at 0.020 mmol/kg, at dose of Dextran or LMWH at 10 mg/kg) for seven times in fourteen days. n =3, Means \pm SD.

Reference:

1. Rydberg HA, Yanez Arteta M, Berg S, Lindfors L, Sigfridsson K. Probing adsorption of DSPE-PEG2000 and DSPE-PEG5000 to the surface of felodipine and griseofulvin nanocrystals. *Int J Pharm* **510**, 232-239 (2016).
2. Yan H, Wei P, Song J, Jia X, Zhang Z. Enhanced anticancer activity in vitro and in vivo of luteolin incorporated into long-circulating micelles based on DSPE-PEG2000 and TPGS. *J Pharm Pharmacol* **68**, 1290-1298 (2016).
3. Zhang J, *et al.* In vitro and in vivo evaluation of folate-mediated PEGylated nanostructured lipid carriers for the efficient delivery of furanodiene. *Drug Dev Ind Pharm* **43**, 1610-1618 (2017).
4. Casper CL, Yamaguchi N, Kiick KL, Rabolt JF. Functionalizing electrospun fibers with biologically relevant macromolecules. *Biomacromolecules* **6**, 1998-2007 (2005).
5. Dick B, Schmidt KG, Eisenmann D, Pfeiffer N. A new method for direct detection of heparin on surface-modified intraocular lenses. A modification of Jaques' toluidine blue staining method. *Ophthalmologica* **211**, 75-78 (1997).
6. Sánchez-Fito MT, Oltra E. Optimized Treatment of Heparinized Blood Fractions to Make Them Suitable for Analysis. *Biopreserv Biobank* **13**, 287-295 (2015).
7. Chen Y, *et al.* A low-molecular-weight heparin-coated doxorubicin-liposome for the prevention of melanoma metastasis. *J Drug Target* **23**, 335-346 (2015).
8. Jiang T, *et al.* Metformin and Docosahexaenoic Acid Hybrid Micelles for Premetastatic Niche Modulation and Tumor Metastasis Suppression. *Nano Lett* **19**, 3548-3562 (2019).
9. De Carlo S, Harris JR. Negative staining and cryo-negative staining of macromolecules and viruses for TEM. *Micron* **42**, 117-131 (2011).
10. Harris JR, Scheffler D. Routine preparation of air-dried negatively stained and unstained specimens on holey carbon support films: a review of applications. *Micron* **33**, 461-480 (2002).

11. Huang JL, *et al.* Lipoprotein-biomimetic nanostructure enables efficient targeting delivery of siRNA to Ras-activated glioblastoma cells via macropinocytosis. *Nat Commun* **8**, 15144 (2017).
12. Pei Y, *et al.* Sequential Targeting TGF- β Signaling and KRAS Mutation Increases Therapeutic Efficacy in Pancreatic Cancer. *Small* **15**, e1900631 (2019).
13. Qiu C, *et al.* Systemic delivery of siRNA by hyaluronan-functionalized calcium phosphate nanoparticles for tumor-targeted therapy. *Nanoscale* **8**, 13033-13044 (2016).
14. Konstantopoulos K, Thomas SN. Cancer cells in transit: the vascular interactions of tumor cells. *Annu Rev Biomed Eng* **11**, 177-202 (2009).
15. Shattil SJ, Kim C, Ginsberg MH. The final steps of integrin activation: the end game. *Nat Rev Mol Cell Biol* **11**, 288-300 (2010).
16. Gay LJ, Felding-Habermann B. Contribution of platelets to tumour metastasis. *Nat Rev Cancer* **11**, 123-134 (2011).
17. Haemmerle M, Stone RL, Menter DG, Afshar-Kharghan V, Sood AK. The Platelet Lifeline to Cancer: Challenges and Opportunities. *Cancer Cell* **33**, 965-983 (2018).
18. Hao C, Xu H, Yu L, Zhang L. Heparin: An essential drug for modern medicine. *Prog Mol Biol Transl Sci* **163**, 1-19 (2019).
19. Weiler JM, Linhardt RJ. Antithrombin III regulates complement activity in vitro. *J Immunol* **146**, 3889-3894 (1991).
20. Ma N, Zhang B, Liu J, Zhang P, Li Z, Luan Y. Green fabricated reduced graphene oxide: evaluation of its application as nano-carrier for pH-sensitive drug delivery. *Int J Pharm* **496**, 984-992 (2015).
21. Zong T, *et al.* Enhanced glioma targeting and penetration by dual-targeting liposome co-modified with T7 and TAT. *J Pharm Sci* **103**, 3891-3901 (2014).
22. Chafe SC, *et al.* Carbonic anhydrase IX promotes myeloid-derived suppressor cell mobilization and establishment of a metastatic niche by stimulating G-CSF production. *Cancer Res* **75**, 996-1008 (2015).

23. Kowanetz M, *et al.* Granulocyte-colony stimulating factor promotes lung metastasis through mobilization of Ly6G+Ly6C+ granulocytes. *Proc Natl Acad Sci U S A* **107**, 21248-21255 (2010).
24. Safarzadeh E, Orangi M, Mohammadi H, Babaie F, Baradaran B. Myeloid-derived suppressor cells: Important contributors to tumor progression and metastasis. *J Cell Physiol* **233**, 3024-3036 (2018).
25. Liu Y, Cao X. Characteristics and Significance of the Pre-metastatic Niche. *Cancer Cell* **30**, 668-681 (2016).
26. Dvorakova M, Landa P. Anti-inflammatory activity of natural stilbenoids: A review. *Pharmacol Res* **124**, 126-145 (2017).
27. Seyed MA, Jantan I, Bukhari SN, Vijayaraghavan K. A Comprehensive Review on the Chemotherapeutic Potential of Piceatannol for Cancer Treatment, with Mechanistic Insights. *J Agric Food Chem* **64**, 725-737 (2016).
28. Wang W, Dong L, Zhao B, Lu J, Zhao Y. E-cadherin is downregulated by microenvironmental changes in pancreatic cancer and induces EMT. *Oncol Rep* **40**, 1641-1649 (2018).
29. Odero-Marah V, Hawsawi O, Henderson V, Sweeney J. Epithelial-Mesenchymal Transition (EMT) and Prostate Cancer. *Adv Exp Med Biol* **1095**, 101-110 (2018).
30. Zhang C, Guo F, Xu G, Ma J, Shao F. STAT3 cooperates with Twist to mediate epithelial-mesenchymal transition in human hepatocellular carcinoma cells. *Oncol Rep* **33**, 1872-1882 (2015).
31. Li Y, *et al.* Inhibitory Effect of Piceatannol on TNF- α -Mediated Inflammation and Insulin Resistance in 3T3-L1 Adipocytes. *J Agric Food Chem* **65**, 4634-4641 (2017).
32. Rijcken E, *et al.* ICAM-1 and VCAM-1 antisense oligonucleotides attenuate in vivo leucocyte adherence and inflammation in rat inflammatory bowel disease. *Gut* **51**, 529-535 (2002).
33. Yan S, *et al.* Clematichinenoside inhibits VCAM-1 and ICAM-1 expression in TNF- α -treated endothelial cells via NADPH oxidase-dependent I κ B kinase/NF- κ B pathway. *Free Radic Biol Med* **78**, 190-201 (2015).

34. Gao J, Wang S, Wang Z. High yield, scalable and remotely drug-loaded neutrophil-derived extracellular vesicles (EVs) for anti-inflammation therapy. *Biomaterials* **135**, 62-73 (2017).
35. Calabriso N, *et al.* Multiple anti-inflammatory and anti-atherosclerotic properties of red wine polyphenolic extracts: differential role of hydroxycinnamic acids, flavonols and stilbenes on endothelial inflammatory gene expression. *Eur J Nutr* **55**, 477-489 (2016).
36. Maeda H. Toward a full understanding of the EPR effect in primary and metastatic tumors as well as issues related to its heterogeneity. *Adv Drug Deliv Rev* **91**, 3-6 (2015).
37. Borsig L, Wong R, Feramisco J, Nadeau DR, Varki NM, Varki A. Heparin and cancer revisited: mechanistic connections involving platelets, P-selectin, carcinoma mucins, and tumor metastasis. *Proc Natl Acad Sci U S A* **98**, 3352-3357 (2001).
38. Ramacciotti E, *et al.* P-selectin/ PSGL-1 inhibitors versus enoxaparin in the resolution of venous thrombosis: a meta-analysis. *Thromb Res* **125**, e138-142 (2010).
39. Mei L, Liu Y, Xia C, Zhou Y, Zhang Z, He Q. Polymer-Drug Nanoparticles Combine Doxorubicin Carrier and Heparin Bioactivity Functionalities for Primary and Metastatic Cancer Treatment. *Mol Pharm* **14**, 513-522 (2017).
40. Anderson RL, *et al.* A framework for the development of effective anti-metastatic agents. *Nat Rev Clin Oncol* **16**, 185-204 (2019).
41. Vanharanta S, Massagué J. Origins of metastatic traits. *Cancer Cell* **24**, 410-421 (2013).
42. Reymond N, D'Água BB, Ridley AJ. Crossing the endothelial barrier during metastasis. *Nature Reviews Cancer* **13**, 858-870 (2013).
43. Rodrigues P, Vanharanta S. Circulating Tumor Cells: Come Together, Right Now, Over Metastasis. *Cancer Discov* **9**, 22-24 (2019).
44. Haemmerle M, Stone RL, Menter DG, Afshar-Kharghan V, Sood AK. The Platelet Lifeline to Cancer: Challenges and Opportunities. *Cancer Cell* **33**, S1535610818300709 (2018).

45. Labelle M, Begum S, Hynes RO. Direct signaling between platelets and cancer cells induces an epithelial-mesenchymal-like transition and promotes metastasis. *Cancer Cell* **20**, 576-590 (2011).
46. Hiratsuka S, *et al.* MMP9 induction by vascular endothelial growth factor receptor-1 is involved in lung-specific metastasis. *Cancer Cell* **2**, 289-300 (2002).
47. Hiratsuka S, *et al.* The S100A8-serum amyloid A3-TLR4 paracrine cascade establishes a pre-metastatic phase. *Nat Cell Biol* **10**, 1349-1355 (2008).
48. Yang J, Weinberg RA. Epithelial-mesenchymal transition: at the crossroads of development and tumor metastasis. *Dev Cell* **14**, 818-829 (2008).
49. Nguyen DX, Bos PD, Massagué J. Metastasis: from dissemination to organ-specific colonization. *Nat Rev Cancer* **9**, 274-284 (2009).
50. Fischer KR, *et al.* Epithelial-to-mesenchymal transition is not required for lung metastasis but contributes to chemoresistance. *Nature* **527**, 472-476 (2015).
51. Castellone DD, Van Cott EM. Laboratory monitoring of new anticoagulants. *Am J Hematol* **85**, 185-187 (2010).
52. Douxfils J, *et al.* Non-VKA Oral Anticoagulants: Accurate Measurement of Plasma Drug Concentrations. *Biomed Res Int* **2015**, 345138 (2015).
53. Lian T, Ho RJ. Trends and developments in liposome drug delivery systems. *J Pharm Sci* **90**, 667-680 (2001).
54. Storm G, Belliot SO, Daemen T, Lasic DD. Surface modification of nanoparticles to oppose uptake by the mononuclear phagocyte system. *Advanced Drug Delivery Reviews* **17**, 31-48 (1995).
55. Bates RJ, Chapman CM, Prout GR, Jr., Lin CW. Immunohistochemical identification of prostatic acid phosphatase: correlation of tumor grade with acid phosphatase distribution. *J Urol* **127**, 574-580 (1982).
56. Uchida T, *et al.* Immunoperoxidase study of alkaline phosphatase in testicular tumor. *Cancer* **48**, 1455-1462 (1981).
57. Bradbury J. Metastasis in colorectal cancer associated with phosphatase expression. *Lancet* **358**, 1245 (2001).

58. Saha S, *et al.* A phosphatase associated with metastasis of colorectal cancer. *Science* **294**, 1343-1346 (2001).
59. Gosset P, *et al.* Expression of E-selectin, ICAM-1 and VCAM-1 on bronchial biopsies from allergic and non-allergic asthmatic patients. *Int Arch Allergy Immunol* **106**, 69-77 (1995).
60. Bailey KA, Moreno E, Haj FG, Simon SI, Passerini AG. Mechanoregulation of p38 activity enhances endoplasmic reticulum stress-mediated inflammation by arterial endothelium. *Faseb j* **33**, 12888-12899 (2019).

REVIEWERS' COMMENTS

Reviewer #1 (Remarks to the Author):

The authors have carefully addressed all my concerns, and I have no more questions for this manuscript.

Reviewer #2 (Remarks to the Author):

This manuscript is improved after addressing referees' comments. However, there are some remaining issues that need to be addressed.

1. In the introduction of background, the authors described so much about the circulating tumor cells (CTCs), but through the whole study, no experimental result was relevant CTCs.
2. In Fig. 1d, the authors should provide TEM images containing at least three NPs to show the typical morphology of CaPP and H@CaPP. Alternatively, such TEM images can be provided in a supplementary figure.
3. In Figure S11, for better comparison, the authors should keep all the graphs with the same axis scale. Also, it may be better to see the difference if the graphs are presented in a stack image, and the peak of PP should be added to better illustrate that piceatannol was finally formed.
4. In the discussion part, the authors should discuss how, when, and where the H@CaPP will release LMWH and the PP could be transformed into the piceatannol.
5. In line 183-185, the authors wrote that "In pH of lysosomes, the H@CaPP released 83.0% of PP in 24 h. Whereas, in pH of lysosomes, only 35.5% of PP was released from the H@CaPP in 24 h (Fig. S10)." This description did not match the result presented in Figure S10. Please go over the whole manuscript and correct similar mistakes. Also, please double check typos and grammar errors.

Reviewer #3 (Remarks to the Author):

- The authors have responded to all, but one, queries (points numeroted 1 to 5) and I have, for these five responses, no other restriction. It has just to be modified one conclusion of the point 5: the authors wrote (lines 520-521): "This result suggested that the nanotherapeutic did not cause side effects on the immune system". They may write: "... on the haematological system".

- The only one lack concerns the issue on the ethics approval for the development of metastases after local surgery, as I have mentioned in my first review: "Hence, because of the high amount of deaths occurring during this experiment, this specific optimization of the in vivo protocol might have been previously approved by an ethics committee, and this approval should be added in the text". I have found the same and general sentence: "Protocols of animal experiments were approved by Institutional Animal Care and Use Committee (IACUC), School of Pharmacy, Fudan University (Shanghai, China)". Can the authors respond to this request?

Point-to-point responses to reviewer's comments

Reviewer #1 (Remarks to the Author):

The authors have carefully addressed all my concerns, and I have no more questions for this manuscript.

Response: Thanks for your suggestions and comments.

Reviewer #2 (Remarks to the Author):

1. In the introduction of background, the authors described so much about the circulating tumor cells (CTCs), but through the whole study, no experimental result was relevant CTCs.

Response: Thank you for the comments. Indeed, it would be optimal to use CTCs for the evaluations. There are some technologies (magnetic beads, microfluidic chips and luminescent nanoprobe) currently available for CTC isolation^{1, 2, 3}. However, since CTCs are extremely scarce in the circulation (as low as 1 CTC per 10⁷ blood cells) and the sensitivity and selectivity of the above technologies are still quite low, it is extremely difficult to detect and capture enough CTCs from the circulation for both the *in vivo* and *in vitro* experiments^{4, 5, 6}. In addition, the phenotype of CTCs relates to the different stage of tumor metastasis^{7, 8, 9}. Hence, instead of directly using CTCs, most previous works used tumor cells to estimate the impact of drug or nanomedicine treatment on CTCs both *in vitro* and *in vivo* (e.g. *Sci Adv.* 2018;4(9): eaat7828, *Proc Natl Acad Sci.* 2014;111(3):930-5 and *Cell.* 2014;159(1):176-187.). Accordingly, we performed the adhesion experiment by using tumor cells (4T1 cells) which overexpress epithelial cell adhesion molecule (EPCAM, as a common CTC biomarker) as the model to evaluate whether the nanotherapeutic could inhibit adhesion of platelets to CTCs, finding that H@CaPP could interfere the aberrant platelets aggregation on the cells^{10, 11}. Moreover, *in vivo* immunohistochemistry assay also showed that H@CaPP could hinder the platelet (labeled with CD-41) aggregation around the tumor cells. Hence, we believe the results supported that H@CaPP could hinder the circulation phase of tumor metastasis via inhibiting the formation of “micro-thrombi”. The relevant results and discussions please see Line 343-386 (Page 16-18) in the revised manuscript.

2. In Fig. 1d, the authors should provide TEM images containing at least three NPs to show the typical morphology of CaPP and H@CaPP. Alternatively, such TEM images can be provided in a supplementary figure.

Response: Thank you for the comment and suggestion. Accordingly, we have provided the TEM images containing more than three nanoparticles in the revised Supplementary Fig. 10. Under TEM, H@CaPP exhibited a dimmer layer that was not

found on CaPP. The difference between CaPP and H@CaPP confirmed the successful adsorption of anionic LMWH onto the surfaces of CaPP.

Supplementary Figure 10 TEM images of CaPP and H@CaPP. Scale bar, 50 nm.

3. In Figure S11, for better comparison, the authors should keep all the graphs with the same axis scale. Also, it may be better to see the difference if the graphs are presented in a stack image, and the peak of PP should be added to better illustrate that piceatannol was finally formed.

Response: Thank you for the comments and suggestions. Accordingly, for clearly indicating the retention times of piceatannol and PP, HPLC spectrums of the standards at high concentrations were shown in the Supplementary Fig. 13b-c, the axis scale in each of the HPLC spectrum (Supplementary Fig. 13b-i) and presented the stack images (Supplementary Fig. 13f & i). The peak of PP could not be detected in the cells culture medium after incubating with H@CaPP because the PP would be transformed to piceatannol in cells. The relevant results and analyses please see Line 190-204 (Page 8) in the revised manuscript.

Supplementary Figure 13 Conversion of PP to piceatannol in 4T1/HUVEC cells. **a** Determination of piceatannol in the cell medium after incubating 4T1/HUVEC cells with H@CaPP at 37°C for 4 h. **b** HPLC spectrum of piceatannol standard. **c** HPLC spectrum of PP standard. **d** HPLC spectrum of the 4T1 cells culture medium. **e** HPLC spectrum of the 4T1 cells culture medium after incubating with H@CaPP at 37°C for 4 h. **f** The stack image of **d** and **e**. **g** HPLC spectrum of the HUVEC cells culture medium. **h** HPLC spectrum of the HUVEC cells culture medium after incubating with H@CaPP at 37°C for 4 h. **i** The stack image of **g** and **h**.

4. In the discussion part, the authors should discuss how, when, and where the H@CaPP will release LMWH and the PP could be transformed into the piceatannol.

Response: Thank you for the comment. LMWH on H@CaPP could bind to the P-selectin on the surface of tumor cells and activated endothelial cells (ECs) in the tumor site and pre-metastatic niche, respectively^{12, 13, 14}. In addition, LMWH on H@CaPP could hinder the formation of “micro-thrombi” via platelets aberrant aggregation on the tumor cells. As to the PP, firstly, the nanoparticles were internalized by the cells. After that, piceatannol phosphate (PP) could be transformed into the piceatannol via cellular phosphatases. The phosphatase exists extensively on cell membranes, cytoplasm, and lysosomes, especially in tumor cells^{15, 16, 17}. In release study, the H@CaPP released 83.0% of PP in 24 h at pH of lysosomes (Supplementary Fig. 12). The result indicated that PP was released in lysosomes. To study the intracellular conversion of PP to piceatannol, we incubated 4T1 and HUVECs cells with H@CaPP for 4 h, and successfully detected piceatannol in the culture media, which indicated that PP could be transformed to the active drug piceatannol via cellular phosphatases in cells (Supplementary Fig. 13). The relevant discussions please see Line 616-629 (Page 28-29) in the revised manuscript.

Supplementary Figure 12 The PP release profile of H@CaPP in different pH medium (5.0 or 6.5) within 24 h (mean \pm SD, n = 3 samples per group). Error bars represent SD.

Supplementary Figure 13 Conversion of PP to piceatannol in 4T1/HUVEC cells. **a** Determination of piceatannol in the cell medium after incubating 4T1/HUVEC cells with H@CaPP at 37°C for 4 h. **b** HPLC spectrum of piceatannol standard. **c** HPLC spectrum of PP standard. **d** HPLC spectrum of the 4T1 cells culture medium. **e** HPLC spectrum of the 4T1 cells culture medium after incubating with H@CaPP at 37°C for 4 h. **f** The stack image of **d** and **e**. **g** HPLC spectrum of the HUVEC cells culture medium. **h** HPLC spectrum of the HUVEC cells culture medium after incubating with H@CaPP at 37°C for 4 h. **i** The stack image of **g** and **h**.

5. In line 183-185, the authors wrote that “In pH of lysosomes, the H@CaPP released 83.0% of PP in 24 h. Whereas, in pH of lysosomes, only 35.5% of PP was released from the H@CaPP in 24 h (Fig. S10).” This description did not match the result presented in Figure S10. Please go over the whole manuscript and correct similar mistakes. Also, please double check typos and grammar errors.

Response: Thank you very much for pointing out the mistake and we have corrected the description accordingly please see Line 188-189 (Page 7) in the revised manuscript. And we have checked the whole manuscript and corrected the editing mistakes.

Reviewer #3 (Remarks to the Author):

- The authors have responded to all, but one, queries (points numeroted 1 to 5) and I have, for these five responses, no other restriction. It has just to be modified one conclusion of the point 5: the authors wrote (lines 520-521): "This result suggested that the nanotherapeutic did not cause side effects on the immune system". They may write: "... on the hematological system".

Response: Thank you for the comment. Accordingly, we have rewritten the conclusion : “This result suggested that the nanotherapeutic did not cause side effects on the hematological system.”.

- The only one lack concerns the issue on the ethics approval for the development of metastases after local surgery, as I have mentioned in my first review: "Hence, because of the high amount of deaths occurring during this experiment, this specific optimization of the in vivo protocol might have been previously approved by an ethics committee, and this approval should be added in the text". I have found the same and general sentence: "Protocols of animal experiments were approved by Institutional Animal Care and Use Committee (IACUC), School of Pharmacy, Fudan University (Shanghai, China)". Can the authors respond to this request?

Response: Thanks for your concern and suggestion. For avoiding the death of mice caused by operation, tumor removal surgery was carried out under the guidance of professional surgeon. According to your suggestion, the “Animal Research Ethical Application” and “Experimental Animal Ethics Committee of School of Pharmacy Fudan University” were uploaded to the submission website, and the detailed protocol for animal experiment (Line 817-842, Page 36) and the ethical approval number (Line 701, Page 31) were added in the revised manuscript.

Reference

1. Sha MY, Xu H, Natan MJ, Cromer R. Surface-enhanced Raman scattering tags for rapid and homogeneous detection of circulating tumor cells in the presence of human whole blood. *J Am Chem Soc* **130**, 17214-17215 (2008).
2. Zhang J, *et al.* DNA Nanolithography Enables a Highly Ordered Recognition Interface in a Microfluidic Chip for the Efficient Capture and Release of Circulating Tumor Cells. *Angew Chem Int Ed Engl* **59**, 14115-14119 (2020).
3. Guo H, *et al.* Direct Detection of Circulating Tumor Cells in Whole Blood Using Time-Resolved Luminescent Lanthanide Nanoprobe. *Angew Chem Int Ed Engl* **58**, 12195-12199 (2019).
4. Cho HY, *et al.* Selective isolation and noninvasive analysis of circulating cancer stem cells through Raman imaging. *Biosens Bioelectron* **102**, 372-382 (2018).
5. Shen Z, Wu A, Chen X. Current detection technologies for circulating tumor cells. *Chem Soc Rev* **46**, 2038-2056 (2017).
6. Mitchell MJ, Wayne E, Rana K, Schaffer CB, King MR. TRAIL-coated leukocytes that kill cancer cells in the circulation. *Proc Natl Acad Sci U S A* **111**, 930-935 (2014).
7. Kuo AH, Clarke MF. Identifying the metastatic seeds of breast cancer. *Nat Biotechnol* **31**, 504-505 (2013).
8. Gires O, Pan M, Schinke H, Canis M, Baeuerle PA. Expression and function of epithelial cell adhesion molecule EpCAM: where are we after 40 years? *Cancer Metastasis Rev* **39**, 969-987 (2020).
9. Austin RG, Huang TJ, Wu M, Armstrong AJ, Zhang T. Clinical utility of non-EpCAM based circulating tumor cell assays. *Adv Drug Deliv Rev* **125**, 132-142 (2018).
10. Alix-Panabières C, Pantel K. Challenges in circulating tumour cell research. *Nature Reviews Cancer* **14**, 623-631 (2014).
11. Wu L, Qu X. Cancer biomarker detection: recent achievements and challenges. *Chem Soc Rev* **44**, 2963-2997 (2015).
12. Amano H, *et al.* Angiotensin II Type 1A Receptor Signaling Facilitates Tumor

Metastasis Formation through P-Selectin–Mediated Interaction of Tumor Cells with Platelets and Endothelial Cells. *The American Journal of Pathology* **182**, 553-564 (2013).

13. Ludwig, R. J. Endothelial P-Selectin as a Target of Heparin Action in Experimental Melanoma Lung Metastasis. *Cancer Research*, (2004).
14. Shigeta A, Matsumoto M, Tedder TF, Lowe JB, Miyasaka M, Hirata T. An L-selectin ligand distinct from P-selectin glycoprotein ligand-1 is expressed on endothelial cells and promotes neutrophil rolling in inflammation. *Blood* **112**, 4915-4923 (2008).
15. Jia-Lin, *et al.* Lipoprotein-biomimetic nanostructure enables efficient targeting delivery of siRNA to Ras-activated glioblastoma cells via macropinocytosis. *Nature Communications*, (2017).
16. Qiu C, *et al.* Systemic delivery of siRNA by hyaluronan-functionalized calcium phosphate nanoparticles for tumor-targeted therapy. *Nanoscale*, 13033-13044 (2016).
17. Yuanyuan, *et al.* Sequential Targeting TGF- β Signaling and KRAS Mutation Increases Therapeutic Efficacy in Pancreatic Cancer. *Small*, (2019).